# STRUCTURE-BASED DRUG DESIGN WITH EQUIVARIANT DIFFUSION MODELS

## ABSTRACT

Structure-based drug design (SBDD) aims to design small-molecule ligands that bind with high affinity and specificity to pre-determined protein targets. Traditional SBDD pipelines start with large-scale docking of compound libraries from public databases, thus limiting the exploration of chemical space to existent previously studied regions. Recent machine learning methods approached this problem using an atom-by-atom generation approach, which is computationally expensive. In this paper, we formulate SBDD as a 3D-conditional generation problem and present DiffSBDD, an SE(3)-equivariant 3D-conditional diffusion model that generates novel ligands conditioned on protein pockets. Furthermore, we curate a new dataset of experimentally determined binding complex data from Binding MOAD to provide a realistic binding scenario that complements the synthetic CrossDocked dataset. Comprehensive in silico experiments demonstrate the efficiency of DiffSBDD in generating novel and diverse drug-like ligands that engage protein pockets with high binding energies as predicted by in silico docking.

## 1 INTRODUCTION

The rational design of molecular compounds to act as drugs remains an oustanding challenge in biopharmaceutical research. Towards supporting such efforts, structure-based drug design (SBDD) aims to generate small-molecule ligands that bind to a specific 3D protein structure with high affinity and specificity (Anderson, 2003). However, SBDD remains very challenging and with important limitations. A traditional SBDD campaign starts with the identification and validation of a target of interest and its subsequent structural characterization using experimental structural determination methods. The first step in this process is the identification of the binding pocket; a cavity in which ligands may bind the target to elicit the desired therapeutic effect. This can be achieved via experimental means or a plethora of computational approaches (Pérot et al., 2010). Once a binding site is identified, the goal is to discover lead compounds that exhibit the desired biological activity. Importantly, to transition from leads to promising candidates the compounds need to be evaluated regarding other drug development constraints that are also hard to predict (toxicity, absorption, etc.).

Traditionally, SBDD is handled either by high-throughput experimental or virtual screening (Lyne, 2002; Shoichet, 2004) of large chemical databases. Not only is this expensive and time consuming but it also limits the exploration of chemical space to the historical knowledge of previously studied molecules, with a further emphasis usually placed on commercial availability (Irwin & Shoichet, 2005). Moreover, the optimization of initial lead molecules is often a biased process, with heavy reliance on human intuition (Ferreira et al., 2015).

Recent advances in geometric deep learning, especially in modeling geometric structures of biomolecules (Bronstein et al., 2021; Atz et al., 2021), provide a promising direction for structure-based drug design (Gaudelet et al., 2021). Even though utilizing deep learning as surrogate docking models has achieved remarkable progress (Lu et al., 2022; Stärk et al., 2022), deep learning-based design of ligands that bind to target proteins is still an open problem. Early attempts have been made to represent molecules as atomic density maps, and variational auto-encoders were utilized to generate new atomic density maps corresponding to novel molecules (Ragoza et al., 2022). However, it is nontrivial to map atomic density maps back to molecules, necessitating a subsequent atom-fitting stage. Follow-up work addressed this limitation by representing molecules as 3D graphs with atomic coordinates and types which circumvents the unnecessary post-processing

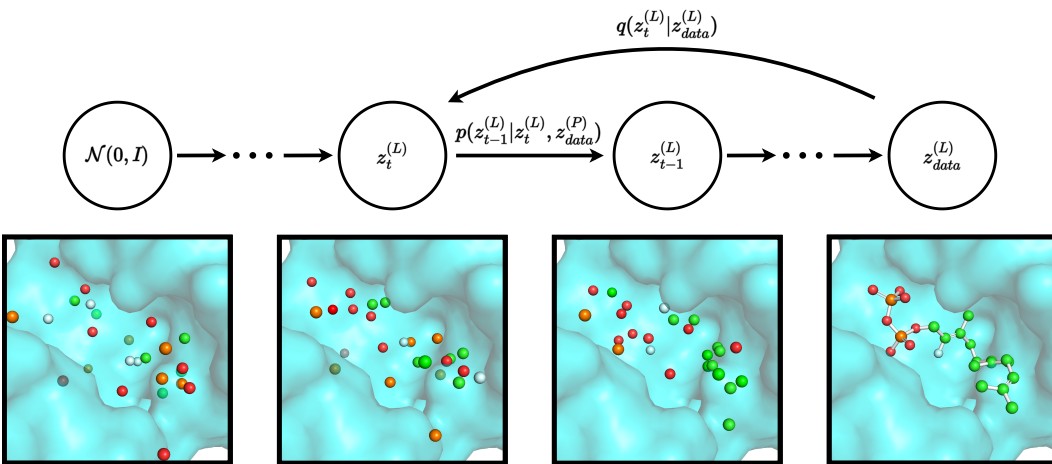

Figure 1: DiffSBDD in the protein-conditioned scenario. We first simulate the forward diffusion process $q$ to gain a trajectory of progressively noised samples over T timesteps. We then train a model $p_\theta$ to reverse or denoise this process that is conditional on the target structure. Once trained, we are able to sample new drug candidates from a Gaussian distribution $\mathcal{N}(\mathbf{0}, \mathbf{I})$. Both atom features and coordinates are diffused throughout the process. Ligands ($z^{(L)}$) are represented as fully-connected graphs during the diffusion process (edges not shown for clarity) and covalent bonds are added to the resultant point cloud at the end of generation. The protein ($z^{(P)}$) is represented as a graph but is shown as a surface here for clarity.

steps. Li et al. (2021) proposed an autoregressive generative model to sample ligands given the protein pocket as a conditioning constraint. Peng et al. (2022) improved this method by using an $E(3)$-equivariant graph neural network which respects rotation and translation symmetries in 3D space. Similarly, Drotár et al. (2021); Liu et al. (2022) used autoregressive models to generate atoms sequentially and incorporate angles during the generation process. Li et al. (2021) formulated the generation process as a reinforcement learning problem and connected the generator with Monte Carlo Tree Search for protein pocket-conditioned ligand generation. However, the main premise of sequential generation methods may not hold in real scenarios, since there is no ordering of the generation process and, as a result, the global context of the generated ligands may be lost. In addition, sequential methods pose more computational complexities that make the model inference inefficient (Luo et al., 2021; Peng et al., 2022).

An alternative is a one-shot generation strategy that samples the atomic coordinates and types of all the atoms at once (Du et al., 2022b). In this work, we develop an equivariant diffusion model for structure-based drug design (DiffSBDD) which, to the best of our knowledge, is the first of its kind. Specifically, we formulate SBDD as a 3D-conditioned generation problem where we aim to generate diverse ligands with high binding affinity for specific protein targets. We propose an $SE(3)$-equivariant 3D-conditional diffusion model that respects translation, rotation, and permutation equivariance. We introduce two strategies, *protein-conditioned generation* and *ligand-inpainting generation* producing new ligands conditioned on protein pockets. Specifically, protein-conditioned generation considers the protein as a fixed context, while ligand-inpainting models the joint distribution of the protein-ligand complex and new ligands are inpainted at inference time. We also demonstrate that our model can be used for out-of-the-box for molecular optimization. We further curate an experimentally determined binding dataset derived from Binding MOAD (Hu et al., 2005), which supplements the commonly used synthetic CrossDocked (Francoeur et al., 2020) dataset to validate our model performance under realistic binding scenarios. The experimental results demonstrate that DiffSBDD is capable of generating novel, diverse and drug-like ligands with predicted high binding affinities to given protein pockets. The code is available at https://anonymous.4open.science/r/DiffSBDD-AF75/.

## 2 BACKGROUND

**Denoising Diffusion Probabilistic Models** Denoising diffusion probabilistic models (DDPMs) (Sohl-Dickstein et al., 2015; Ho et al., 2020) are a class of generative models in-

spired by non-equilibrium thermodynamics. Briefly, they define a Markovian chain of random diffusion steps by slowly adding noise to sample data and then learning the reverse of this process (typically via a neural network) to reconstruct data samples from noise.

In this work, we closely follow the framework developed by Hoogeboom et al. (2022). In our setting, data samples are atomic point clouds $\boldsymbol{z}_{\text{data}} = [\boldsymbol{x}, \boldsymbol{h}]$ with 3D geometric coordinates $\boldsymbol{x} \in \mathbb{R}^{N \times 3}$ and categorical features $\boldsymbol{h} \in \mathbb{R}^{N \times d}$, where $N$ is the number of atoms. A fixed noise process

$$q(\boldsymbol{z}_t | \boldsymbol{z}_{\text{data}}) = \mathcal{N}(\boldsymbol{z}_t | \alpha_t \boldsymbol{z}_{\text{data}}, \sigma_t^2 \boldsymbol{I}) \tag{1}$$

adds noise to the data $\boldsymbol{z}_{\text{data}}$ and produces a latent noised representation $\boldsymbol{z}_t$ for $t = 0, \dots, T$. $\alpha_t$ controls the signal-to-noise ratio $\text{SNR}(t) = \alpha_t^2 / \sigma_t^2$ and follows either a learned or pre-defined schedule from $\alpha_0 \approx 1$ to $\alpha_T \approx 0$ (Kingma et al., 2021). We also choose a variance-preserving noising process (Song et al., 2020) with $\alpha_t = \sqrt{1 - \sigma_t^2}$.

Since the noising process is Markovian, we can write the denoising transition from time step $t$ to $s < t$ in closed form as

$$q(\boldsymbol{z_s} | \boldsymbol{z}_{\text{data}}, \boldsymbol{z}_t) = \mathcal{N}\left(\boldsymbol{z_s} \Big| \frac{\alpha_{t|s} \sigma_s^2}{\sigma_t^2} \boldsymbol{z}_t + \frac{\alpha_s \sigma_{t|s}^2}{\sigma_t^2} \boldsymbol{z}_{\text{data}}, \frac{\sigma_{t|s}^2 \sigma_s^2}{\sigma_t^2} \boldsymbol{I}\right) \tag{2}$$

with $\alpha_{t|s} = \frac{\alpha_t}{\alpha_s}$ and $\sigma_{t|s}^2 = \sigma_t^2 - \alpha_{t|s}^2 \sigma_s^2$ following the notation of Hoogeboom et al. (2022). This true denoising process depends on the data sample $\boldsymbol{z}_{\text{data}}$, which is not available when using the model for generating new samples. Instead, a neural network $\phi_\theta$ is used to approximate the sample $\hat{\boldsymbol{z}}_{\text{data}}$. More specifically, we can reparameterize Equation (1) as $\boldsymbol{z}_t = \alpha_t \boldsymbol{z}_{\text{data}} + \sigma_t \boldsymbol{\epsilon}$ with $\boldsymbol{\epsilon} \sim \mathcal{N}(\boldsymbol{0}, \boldsymbol{I})$ and directly predict the Gaussian noise $\hat{\boldsymbol{\epsilon}}_\theta = \phi_\theta(\boldsymbol{z}_t, t)$. Thus, $\hat{\boldsymbol{z}}_{\text{data}}$ is simply given as $\hat{\boldsymbol{z}}_{\text{data}} = \frac{1}{\alpha_t} \boldsymbol{z}_t - \frac{\sigma_t}{\alpha_t} \hat{\boldsymbol{\epsilon}}_\theta$.

The neural network is trained to maximise the likelihood of observed data by optimising a variational lower bound on the data, which is equivalent to the simplified training objective (Ho et al., 2020; Kingma et al., 2021) $\mathcal{L}_{\text{train}} = \frac{1}{2} || \boldsymbol{\epsilon} - \phi_\theta(\boldsymbol{z}_t, t) ||^2$ up to a scale factor (see Appendix A for details).

$E(n)$**-equivariant Graph Neural Networks** A function $f : \mathcal{X} \to \mathcal{Y}$ is said to be *equivariant* w.r.t. the group $G$ if $f(g.\boldsymbol{x}) = g.f(\boldsymbol{x})$, where $g.$ denotes the action of the group element $g \in G$ on $\mathcal{X}$ and $\mathcal{Y}$ (Serre et al., 1977). Graph Neural Networks (GNNs) are learnable functions that process graph-structured data in a permutation-equivariant way, making them particularly useful for molecular systems where nodes do not have an intrinsic order. Permutation invariance means that $\text{GNN}(\boldsymbol{\Pi}\mathbf{X}) = \boldsymbol{\Pi}\,\text{GNN}(\mathbf{X})$ where $\boldsymbol{\Pi} \in \Sigma_n$ is an $n \times n$ permutation matrix acting on the node feature matrix. Since the nodes of the molecular graph represent the 3D coordinates of atoms, we are interested in additional equivariance w.r.t. the Euclidean group $E(3)$ or rigid transformations. An $E(3)$-equivariant GNN (EGNN) satisfies $\text{EGNN}(\boldsymbol{\Pi}\mathbf{X}\mathbf{A} + \mathbf{b}) = \boldsymbol{\Pi}\,\text{EGNN}(\mathbf{X})\mathbf{A} + \mathbf{b}$ for an orthogonal $3 \times 3$ matrix $\mathbf{A}^\top \mathbf{A} = \mathbf{I}$ and some translation vector $\mathbf{b}$ added row-wise.

In our case, since the nodes have both geometric atomic coordinates $\boldsymbol{x}$ as well as atomic type features $\boldsymbol{h}$, we can use a simple implementation of EGNN proposed by Satorras et al. (2021), in which the updates for features $\boldsymbol{h}$ and coordinates $\boldsymbol{x}$ of node $i$ at layer $l$ are computed as follows:

$$\boldsymbol{m}_{ij} = \phi_e(\boldsymbol{h}_i^l, \boldsymbol{h}_j^l, d_{ij}^2, a_{ij}), \; \tilde{e}_{ij} = \phi_{\text{att}}(\boldsymbol{m}_{ij}) \tag{3}$$

$$\boldsymbol{h}_i^{l+1} = \phi_h(\boldsymbol{h}_i^l, \sum_{j \neq i} \tilde{e}_{ij} \boldsymbol{m}_{ij}) \tag{4}$$

$$\boldsymbol{x}_i^{l+1} = \boldsymbol{x}_i^l + \sum_{j \neq i} \frac{\boldsymbol{x}_i^l - \boldsymbol{x}_j^l}{d_{ij} + 1} \phi_x(\boldsymbol{h}_i^l, \boldsymbol{h}_j^l, d_{ij}^2, a_{ij}) \tag{5}$$

where $\phi_e$, $\phi_{\text{att}}$, $\phi_h$ and $\phi_h$ are learnable Multi-layer Perceptrons (MLPs) and $d_{ij}$ and $a_{ij}$ are the relative distances and edge features between nodes $i$ and $j$ respectively.

## 3 EQUIVARIANT DIFFUSION MODELS FOR SBDD

We utilize an equivariant DDPM to generate molecules and binding conformations jointly with respect to a specific protein target. We represent protein and ligand point clouds as fully-connected graphs that are further processed by EGNNs (Satorras et al., 2021). We consider two distinct approaches to 3D pocket conditioning: (1) a conditional DDPM that receives a fixed pocket representation as context in each denoising step, and (2) a model that approximates the joint distribution of ligand-pocket pairs combined with inpainting at inference time.

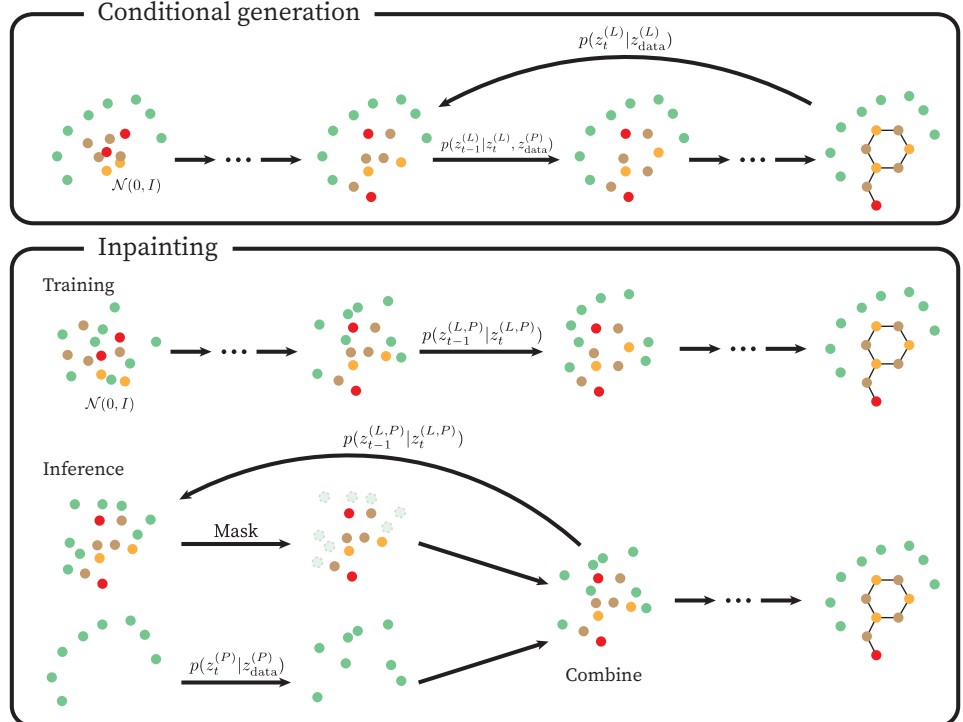

Figure 2: Comparison between the conditional generation and inpainting approaches. The conditional model learns to denoise molecules $\boldsymbol{z}^{(L)}$ in the fixed context of protein pockets $\boldsymbol{z}_{\text{data}}^{(P)}$. In the inpainting scenario, the model first learns to approximate the joint distribution of ligand and pocket nodes $\boldsymbol{z}_{\text{data}}^{(L,P)}$. For sampling, context is provided by combining the latent representation of the ligand with a forward diffused representation of the pocket in each denoising step.

### 3.1 POCKET-CONDITIONED SMALL MOLECULE GENERATION

In the conditional molecule generation setup, we provide fixed three-dimensional context in each step of the denoising process. To this end, we supplement the ligand node point cloud $\boldsymbol{z}_t^{(L)}$, denoted by superscript $L$, with protein pocket nodes $\boldsymbol{z}_{\text{data}}^{(P)}$, denoted by superscript $P$, that remain unchanged throughout the reverse diffusion process (Figure 2).

We parameterize the noise predictor $\hat{\boldsymbol{\epsilon}}_\theta = \phi_\theta(\boldsymbol{z}_t^{(L)}, \boldsymbol{z}_{\text{data}}^{(P)}, t)$ with an EGNN (Satorras et al., 2021; Hoogeboom et al., 2022). To process ligand and pocket nodes with a single GNN, atom types and residue types are first embedded in a joint node embedding space by separate learnable MLPs. We employ the same message-passing scheme outlined in Equations (3)-(5), however, following (Igashov et al., 2022) we do not update the coordinates of nodes that belong to the pocket to ensure the three-dimensional protein context remains fixed throughout the EGNN layers.

**Equivariance** In the probabilistic setting with 3D-conditioning, we would like to ensure $SE(3)$-equivariance in the following sense. This definition explicitly excludes reflections which are connected with chirality and can alter a biomolecule's properties.[1]:

- Evaluating the likelihood of a molecule $\boldsymbol{x}^{(L)} \in \mathbb{R}^{3 \times N_L}$ given the three-dimensional representation of a protein pocket $\boldsymbol{x}^{(P)} \in \mathbb{R}^{3 \times N_P}$ should not depend on global $SE(3)$-transformations of the system, i.e. $p(\boldsymbol{R}\boldsymbol{x}^{(L)}+\boldsymbol{t}|\boldsymbol{R}\boldsymbol{x}^{(P)}+\boldsymbol{t}) = p(\boldsymbol{x}^{(L)}|\boldsymbol{x}^{(P)})$ for orthogonal $\boldsymbol{R} \in \mathbb{R}^{3 \times 3}$ with $\boldsymbol{R}^T\boldsymbol{R} = \boldsymbol{I}$, $\det(\boldsymbol{R}) = 1$ and $\boldsymbol{t} \in \mathbb{R}^3$ added column-wise.

- At the same time, it should be possible to generate samples $\boldsymbol{x}^{(L)} \sim p(\boldsymbol{x}^{(L)}|\boldsymbol{x}^{(P)})$ from this conditional probability distribution so that equivalently transformed ligands $\boldsymbol{R}\boldsymbol{x}^{(L)}+\boldsymbol{t}$

---

[1]We transpose the node feature matrices hereafter so that the matrix multiplication resembles application of a group action. We also ignore node type features, which transform invariantly, for simpler notation.

are sampled with the same probability if the input pocket is rotated and translated and we sample from $p(\boldsymbol{R}\boldsymbol{x}^{(L)} + \boldsymbol{t}|\boldsymbol{R}\boldsymbol{x}^{(P)} + \boldsymbol{t})$.

Equivariance to the orthogonal group $O(3)$ (comprising rotations and reflections) is achieved because we model both prior and transition probabilities with isotropic Gaussians where the mean vector transforms equivariantly w.r.t. rotations of the context (see Hoogeboom et al. (2022) and Appendix E). Ensuring translation equivariance, however, is not as easy because the transition probabilities $p(\boldsymbol{z}_{t-1}|\boldsymbol{z}_t)$ are not inherently translation-equivariant. In order to circumvent this issue, we follow previous works (Köhler et al., 2020; Xu et al., 2022; Hoogeboom et al., 2022) by limiting the whole sampling process to a linear subspace where the center of mass (CoM) of the system is zero. In practice, this is achieved by subtracting the center of mass of the system before performing likelihood computations or denoising steps. Since equivariance of the transition probabilities depends on the parameterization of the noise predictor $\hat{\boldsymbol{\epsilon}}_\theta$, we can make the model sensitive to reflections with a simple additive term in the EGNN's coordinate update:

$$\boldsymbol{x}_i^{l+1} = \boldsymbol{x}_i^l + \sum_{j \neq i} \frac{\boldsymbol{x}_i^l - \boldsymbol{x}_j^l}{d_{ij} + 1} \phi_x^d(\boldsymbol{h}_i^l, \boldsymbol{h}_j^l, d_{ij}^2, a_{ij}) + \frac{(\boldsymbol{x}_i^l - \bar{\boldsymbol{x}}^l) \times (\boldsymbol{x}_j^l - \bar{\boldsymbol{x}}^l)}{||(\boldsymbol{x}_i^l - \bar{\boldsymbol{x}}^l) \times (\boldsymbol{x}_j^l - \bar{\boldsymbol{x}}^l)|| + 1} \phi_x^\times(\boldsymbol{h}_i^l, \boldsymbol{h}_j^l, d_{ij}^2, a_{ij}),$$

(6)

using the cross product which changes sign under reflection. Here, $\bar{\boldsymbol{x}}^l$ denotes the center of mass of all nodes at layer $l$. $\phi_x^\times$ is an additional MLP. This modification is discussed in more detail in Appendix F.

## 3.2 JOINT DISTRIBUTION WITH INPAINTING

As an extension to the conditional approach described above, we also present a ligand-*inpainting* approach. Originally introduced as a technique for completing masked parts of images (Song et al., 2020; Lugmayr et al., 2022), inpainting has been adopted in other domains, including biomolecular structures (Wang et al., 2022). Here, we extend this idea to three-dimensional point cloud data.

We first train an unconditional DDPM to approximate the joint distribution of ligand and pocket nodes $p(\boldsymbol{z}_{\text{data}}^{(L)}, \boldsymbol{z}_{\text{data}}^{(P)})^2$. This allows us to sample new pairs without additional context. To condition on a target protein pocket, we then need to inject context into the sampling process by modifying the probabilistic transition steps. The combined latent representation $\boldsymbol{z}_{t-1}^{(L,P)}$ of protein pocket and ligand at diffusion step $t - 1$ is assembled from a forward noised version of the pocket that is combined with ligand nodes predicted by the DDPM based on the previous latent representation at step $t$

$$\boldsymbol{z}_{t-1,\text{known}}^{(P)} \sim p(\boldsymbol{z}_{t-1}^{(P)}|\boldsymbol{z}_{\text{data}}^{(P)}) \tag{7}$$

$$\boldsymbol{z}_{t-1,\text{unknown}}^{(L,P)} \sim p_\theta(\boldsymbol{z}_{t-1}^{(L,P)}|\boldsymbol{z}_t^{(L,P)}) \tag{8}$$

$$\boldsymbol{z}_{t-1}^{(L,P)} = \left[\boldsymbol{z}_{t-1,\text{unknown}}^{(L)}, \boldsymbol{z}_{t-1,\text{known}}^{(P)}\right]. \tag{9}$$

In this manner, we traverse the Markov chain in reverse order from $t = T$ to $t = 0$, replacing the predicted pocket nodes with their forward noised counterparts in each step. Equation (8) conditions the generative process on the given protein pocket. Thanks to the noise schedule, which decreases the variance of the noising process to almost zero at $t = 0$ (Equation (1)), the final sample is guaranteed to contain an unperturbed representation of the protein pocket.

Since the model is trained to approximate the unconditional joint distribution of ligand-pocket pairs, the training procedure is identical to the unconditional molecule generation procedure developed by Hoogeboom et al. (2022) aside from the fully-connected neural networks that embed protein and ligand node features in a common space as described in Section 3.1. The conditioning on known protein pockets is entirely delegated to the sampling algorithm, which means this approach is not limited to ligand-inpainting but, in principle, allows us to mask and replace arbitrary parts of the ligand-pocket system without retraining.

Trippe et al. (2022) show that this simple *replacement method* inevitably introduces approximation error that can lead to inconsistent inpainted regions. In our experiments, we observe that the in-

---

[2]We use notations $\boldsymbol{z}^{(L,P)}$ and $[\boldsymbol{z}^{(L)}, \boldsymbol{z}^{(P)}]$ interchangeably to describe the combined system of ligand and pocket nodes.

Table 1: Evaluation of generated molecules for targets from the CrossDocked test set. * denotes that we re-evaluate the generated ligands provided by the authors. The inference times are taken from their papers. Note that these results have been produced with the E(3)-equivariant version of our model and will be updated.

| | Vina Score (kcal/mol, ↓) | QED (↑) | SA (↑) | Lipinski (↑) | Diversity (↑) | Time (s, ↓) |
|---|---|---|---|---|---|---|
| Test set | $-6.871 \pm 2.32$ | $0.476 \pm 0.20$ | $0.728 \pm 0.14$ | $4.340 \pm 1.14$ | — | — |
| 3D-SBDD (AR) (Luo et al., 2021)* | $-5.888 \pm 1.91$ | $0.502 \pm 0.17$ | $0.675 \pm 0.14$ | $4.787 \pm 0.51$ | $0.742 \pm 0.09$ | $19659 \pm 14704$ |
| Pocket2Mol (Peng et al., 2022)* | $-7.058 \pm 2.80$ | $0.572 \pm 0.16$ | $0.752 \pm 0.12$ | $4.936 \pm 0.27$ | $0.735 \pm 0.15$ | $2504 \pm 2207$ |
| DiffSBDD-cond ($C_\alpha$) | $-5.540 \pm 1.57$ | $0.460 \pm 0.14$ | $0.357 \pm 0.09$ | $4.821 \pm 0.45$ | $0.815 \pm 0.06$ | $324 \pm 189$ |
| DiffSBDD-inpaint ($C_\alpha$) | $-5.735 \pm 1.80$ | $0.427 \pm 0.15$ | $0.343 \pm 0.09$ | $4.789 \pm 0.49$ | $0.807 \pm 0.07$ | $329 \pm 177$ |
| DiffSBDD-cond | $-6.584 \pm 2.06$ | $0.495 \pm 0.15$ | $0.336 \pm 0.09$ | $4.795 \pm 0.49$ | $0.730 \pm 0.11$ | $1634 \pm 769$ |

painting solution sometimes generates dislocated molecules that are not properly positioned in the target pocket. Trippe et al. (2022) propose to address this limitation with a particle filtering scheme that upweights more consistent samples in each denoising step. We, however, choose to adopt the conceptually simpler idea of *resampling* (Lugmayr et al., 2022), where each latent representation is repeatedly diffused back and forth before advancing to the next time step. This enables the model to harmonize its prediction for the unknown region and the noisy sample from the known region (Eq. (7)), which does not include any information about the generated part. We choose $r = 10$ resamplings per denoising step based on empirical results discussed in Appendix C.1.

**Equivariance** Similar desiderata as in the conditional case apply to the joint probability model, where we desire $SE(3)$-invariance that can be obtained from invariant priors via equivariant flows (Köhler et al., 2020). The main complications compared to the previous approach are the missing reference frame and impossibility of defining a valid translation-*invariant* prior noise distribution $p(z_T)$ as such a distribution cannot integrate to one. Consequently, it is necessary to restrict the probabilistic model to a CoM-free subspace as described in previous works (Köhler et al., 2020; Xu et al., 2022; Hoogeboom et al., 2022). While the reverse diffusion process is defined for a CoM-free system, substituting the predicted pocket node coordinates with a new diffused version of the known pocket as described in Equations (7) - (9) can lead to non-zero CoM. To prevent this, we translate the known pocket representation so that its center of mass coincides with the predicted representation: $\tilde{x}^{(P)}_{t-1,\text{known}} = x^{(P)}_{t-1,\text{unknown}} - x^{(P)}_{t-1,\text{known}}$ before creating the new combined representation $z^{(L,P)}_{t-1} = [z^{(L)}_{t-1,\text{unknown}}, \tilde{z}^{(P)}_{t-1,\text{known}}]$ with $\tilde{z}^{(P)}_{t-1,\text{known}} = [\tilde{x}^{(P)}_{t-1,\text{known}}, h^{(P)}_{t-1,\text{known}}]$.

## 4 EXPERIMENTS

### 4.1 DATASETS

**CrossDocked** We use the CrossDocked dataset (Francoeur et al., 2020) and follow the same filtering and splitting strategies as in previous work (Luo et al., 2021; Peng et al., 2022). This results in 100,000 high-quality protein-ligand pairs for the training set and 100 proteins for the test set. The split is done by 30% sequence identity using MMseqs2 (Steinegger & Söding, 2017).

**Binding MOAD** We also evaluate our method on experimentally determined protein-ligand complexes found in Binding MOAD (Hu et al., 2005) which are filtered and split based on the proteins' enzyme commission number as described in Appendix D. This results in 40,354 protein-ligand pairs for training and 130 pairs for testing.

### 4.2 EVALUATION

For every experiment, we evaluated all combinations of all-atom and $C_\alpha$ level graphs with conditional and inpainting-based approaches respectively (with the exception of the all-atom inpainting approach due to computational limitations). Full details of model architecture and hyperparameters are given in Appendix C. We sampled 100 valid molecules[3] for each target pocket and removed all atoms that are not bonded to the largest connected fragment. Ligand sizes were sampled from

---

[3]Due to occasional processing issues the actual number of available molecules is slightly lower on average (see Appendix G.1).

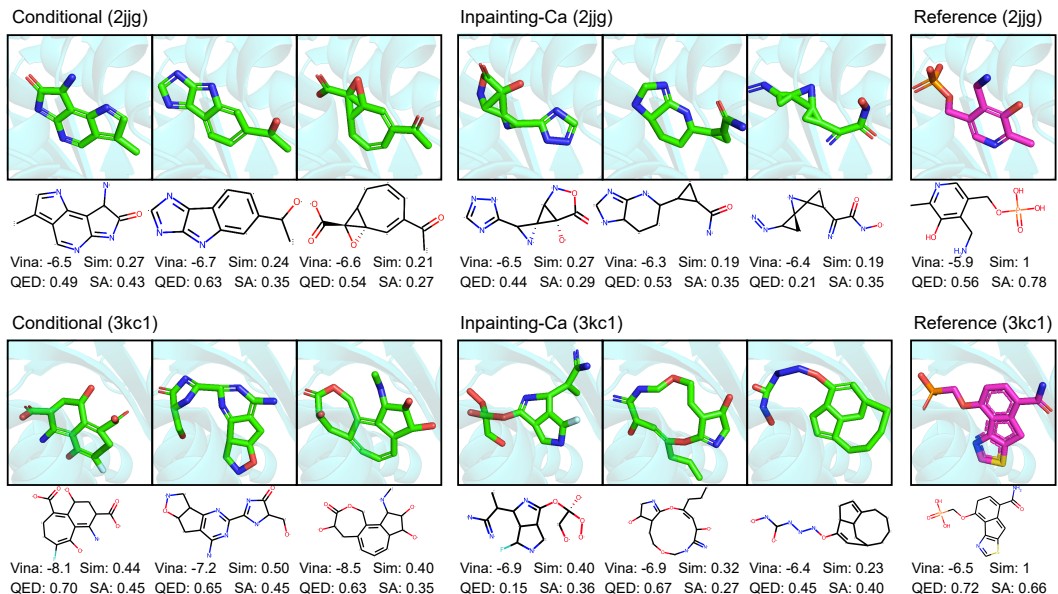

Figure 3: DiffSBDD models trained on CrossDocked and evaluated against a aminotransferase (top, PDB: 2jjg) and hydrolase (bottom, PDB: 3kc1). Conditional and inpainting approaches are compared (using all-atom and $C_\alpha$ level protein presentations respectively) and three high affinity molecules from each model are presented. 'Sim' is the Tanimoto similarity between the generated and reference ligand.

the training distribution as described in Appendix C. This procedure yields significantly smaller molecules than the reference ligands from the test set. We therefore increase the mean ligand size by 15 atoms to sample approximately equally sized molecules. This correction improves the observed docking scores which are highly correlated with the ligand size (see Figure 6).

We employ widely-used metrics to assess the quality of our generated molecules (Peng et al., 2022; Li et al., 2021): (1) **Vina Score** is a physics-based estimation of binding affinity between small molecules and their target pocket; (2) **QED** is a simple quantitative estimation of drug-likeness combining several desirable molecular properties; (3) **SA** (synthetic accessibility) is a measure estimating the difficulty of synthesis; (4) **Lipinski** measures how many rules in the Lipinski rule of five (Lipinski et al., 2012), which is a loose rule of thumb to assess the drug-likeness of molecules, are satisfied; (5) **Diversity** is computed as the average pairwise dissimilarity (1 - *Tanimoto similarity*) between all generated molecules for each pocket; (6) **Inference Time** is the average time to sample 100 molecules for one pocket across all targets. All docking scores and chemical properties are calculated with QuickVina2 (Alhossary et al., 2015) and RDKit (Landrum et al., 2016).

### 4.3 BASELINES

We compare with two recent deep learning methods for structure-based drug design. *3D-SBDD* (Luo et al., 2021) and Pocket2Mol (Peng et al., 2022) are auto-regressive schemes relying on graph representations of the protein pocket and previously placed atoms to predict probabilities based on which new atoms are added. *3D-SBDD* use heuristics to infer bonds from generated atomic point clouds while *Pocket2Mol* directly predicts them during the sequential generation process.

### 4.4 RESULTS

**CrossDocked**   Overall, the experimental results in Table 1 suggest that DiffSBDD can generate diverse small-molecule compounds with predicted high binding affinity, matching state-of-the-art performance. We do not see significant differences between the conditional model and the inpainting approach. The diversity score is arguably the most interesting, as this suggests our model is able to sample greater amounts of chemical space when compared to previous methods, while maintaining

Table 2: Evaluation of generated molecules for target pockets from the Binding MOAD test set.

| | Vina Score (kcal/mol, ↓) | QED (↑) | SA (↑) | Lipinski (↑) | Diversity (↑) | Time (s, ↓) |
|---|---|---|---|---|---|---|
| Test set | $-8.328 \pm 2.05$ | $0.602 \pm 0.15$ | $0.336 \pm 0.08$ | $4.838 \pm 0.37$ | — | — |
| DiffSBDD-cond ($C_\alpha$) | $-6.281 \pm 1.81$ | $0.486 \pm 0.17$ | $0.313 \pm 0.09$ | $4.637 \pm 0.63$ | $0.730 \pm 0.04$ | $44.022 \pm 8.98$ |
| DiffSBDD-inpaint ($C_\alpha$) | $-6.406 \pm 5.13$ | $0.512 \pm 0.17$ | $0.308 \pm 0.09$ | $4.681 \pm 0.58$ | $0.621 \pm 0.16$ | $98.439 \pm 30.44$ |
| DiffSBDD-cond | $-6.726 \pm 1.60$ | $0.470 \pm 0.18$ | $0.331 \pm 0.08$ | $4.666 \pm 0.62$ | $0.711 \pm 0.08$ | $194.860 \pm 49.63$ |

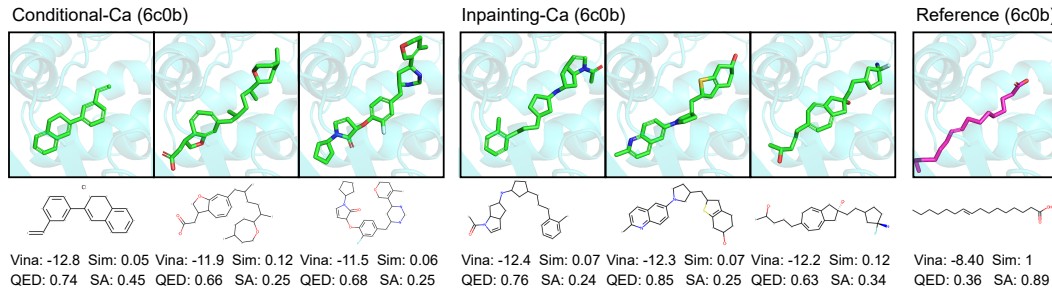

Conditional-Ca (6c0b)     Inpainting-Ca (6c0b)     Reference (6c0b)

Vina: -12.8  Sim: 0.05     Vina: -11.9  Sim: 0.12     Vina: -11.5  Sim: 0.06     Vina: -12.4  Sim: 0.07     Vina: -12.3  Sim: 0.07     Vina: -12.2  Sim: 0.12     Vina: -8.40  Sim: 1
QED: 0.74  SA: 0.45     QED: 0.66  SA: 0.25     QED: 0.68  SA: 0.25     QED: 0.76  SA: 0.24     QED: 0.85  SA: 0.25     QED: 0.63  SA: 0.34     QED: 0.36  SA: 0.89

Figure 4: DiffSBDD models trained on Binding MOAD evaluated against a human receptor protein (PDB: 6c0b). Conditional and inpainting approaches are compared ($C_\alpha$ for both) and the three highest affinity molecules from each model are presented. Further details of the molecules shown here are explained in Appendix G.1

high binding performance, one of the most important requirements in early-stage, structure-based lead discovery. Specifically, DiffSBDD aims to generate ligands that bind to protein pockets and learn the probability density of ligands interacting with protein pockets. While it does not optimize for other molecular properties, such as QED and Lipinski, it generates molecules similar to the test set distributions. Only SA scores are significantly lower on average. While it is unclear why the model fails to approximate the distribution of synthetic accessibility scores successfully, simple techniques can be used for downstream optimization of this property once promising candidates are found (Section 4.5). Generally, presenting the full atomic context to the model constrains the space of outputs considerably, leading to higher Vina scores but lower diversity compared to the $C_\alpha$-only models. The all-atom model consistently beats $C_\alpha$-based models on a per target basis (Appendix Figure 13).

A representative selection of molecules for two targets (*2jjg* and *3kc1*) are presented (Figure 3). This set is curated to be representative of our high scoring molecules, with both realistic and non-realistic motifs shown. It is noteworthy that the second molecule generated for 3kc1 has a similar tricyclic motif in the same pocket location as the reference ligand which was designed by traditional SBDD methods to maximise the hydrophobic interactions via shape complementarity of the ring system (Tsukada et al., 2010). However, a number of irregularities are present in even the highest scoring of generated molecules. For example, the high number of triangles in the molecules targeting 2jjg (from Inpainting-$C_\alpha$) and the large rings for 3kc1 would prove difficult to synthesise. Random selections of generated molecules made by all methods evaluated are presented in Figure 11.

All docking scores reported in Table 1 are within one standard deviation of each other, which poses challenges for the discrimination of the best models. To verify successful pocket-conditioning, we therefore discuss the agreement of generated molecular conformations with poses after docking in Appendix G.5. This experiment showcases the success of our method to model protein-drug interactions at the atomic level and clearly highlights the benefits of the all-atom pocket representation.

**Binding MOAD** Results for the Binding MOAD dataset with experimentally determined binding complex data are reported in Table 2. 100 valid ligands have been generated for each of the 130 test pockets resulting in $13\,000$ molecules in total[4]. DiffSBDD generates highly diverse molecules but on average docking scores are lower than corresponding reference ligands from this dataset.

---

[4]The QuickVina score could not be computed for 49 ($\approx 0.4\%$) molecules from DiffSBDD-cond.

Generated molecules for a representative target are shown in Figure 4. The target (PDB: 6c0b) is a human receptor which is involved in microbial infection (Chen et al., 2018) and possibly tumor suppression (Ding et al., 2016). The reference molecule, a long fatty acid (see Figure 4) that aids receptor binding (Chen et al., 2018), has too high a number of rotatable bonds and low a number of hydrogen bond donors/acceptors to be considered a suitable drug (QED of 0.36). Our model however, generates drug-like (QED between 0.63-0.85) and suitably sized molecules by adding aromatic rings connected by a small number of rotatable bonds, which allows the molecules to adopt a complementary binding geometry and is entropically favourable (by reducing the degrees of freedom), a classic technique in medicinal chemistry (Ritchie & Macdonald, 2009). A random selection of generated molecules in presented in Figure 12.

## 4.5 MOLECULE OPTIMIZATION

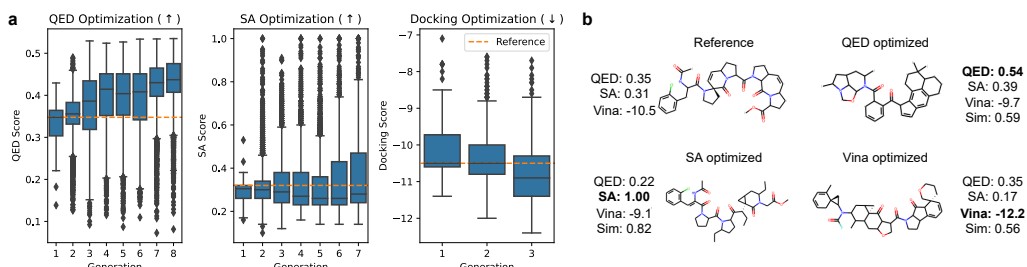

Figure 5: (a) Optimizing for various properties. (b) Examples of optimized molecules.

We use our model to optimize exciting candidate molecules, a common task in drug discovery called lead optimization. This is when we take a compound found to have high binding affinity and optimize it for better 'drug-like' properties. We first noise the atom features and coordinates for $t$ steps (where $t$ is small) using the forward diffusion process. From this partially noised sample, we can then denoise the appropriate number of steps with the reverse process until $t = 0$. This allows us to sample new candidates of various properties whilst staying in the same region of active chemical space, assuming $t$ is small (Appendix Figure 8). This approach is inspired by (Luo et al., 2022) but note this does not allow for direct optimization of specific properties, rather directed exploration around the local chemical space according to what was learnt from the training distribution.

We extend this idea by combining the partial noising/denoising procedure with a simple evolutionary algorithm that optimizes for specific molecular properties. We find that our model performs well at this task out-of-the-box without any additional fine-tuning. As a showcase, we optimize a molecule in the test set targeting PDB:5ndu, a cancer therapeutic (Barone et al., 2020), which has low SA and QED scores, 0.31 and 0.35 respectively, but high binding affinity. Over a number of rounds of optimization, we can observe significant increases in QED (from 0.35 to mean of 0.43) whilst still maintaining high similarity to the original molecule (Figure 5a). We can also rescue the low synthetic accessibility score of the seed molecule by producing a battery of highly accessible molecules when selecting for SA. Finally, we observe that we can perform significant optimization of binding affinity after only a view rounds of optimization. Figure 5b shows 3 representative molecules with substantially optimized scores (QED, SA or Vina) whilst maintaining comparable binding affinity and globally similar structures. Full details are provided in Appendix 4.5.

## 5 CONCLUSION

In this work, we propose DiffSBDD, an $SE(3)$-equivariant 3D-conditional diffusion model for structure-based drug design. We demonstrate the effectiveness and efficiency of DiffSBDD in generating novel and diverse ligands with predicted high-affinity for given protein pockets on both a synthetic benchmark and a new dataset of experimentally determined protein-ligand complexes. We demonstrate that an inpainting-based approach can achieve competitive results to direct conditioning on a wide range of molecular metrics. Extending this more versatile strategy to an all atom pocket representation therefore holds promise to solve a variety of other structure-based drug design tasks, such as lead optimization or linker design, and binding site design without retraining.

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

# Appendix for
# "Structure-based Drug Design with Equivariant Diffusion Models"

## A  VARIATIONAL LOWER BOUND

To maximise the likelihood of our training data, we aim at optimising the variational lower bound (VLB) (Kingma et al., 2021; Hoogeboom et al., 2022)

$$-\log p(\boldsymbol{z}_{\text{data}}) \leq \underbrace{D_{\text{KL}}\big(q(\boldsymbol{z}_T|\boldsymbol{z}_{\text{data}})||p(\boldsymbol{z}_T)\big)}_{\text{prior loss } \mathcal{L}_{\text{prior}}} \underbrace{-\mathbb{E}_{q(\boldsymbol{z}_0|\boldsymbol{z}_{\text{data}})}\big[\log p(\boldsymbol{z}_{\text{data}}|\boldsymbol{z}_0)\big]}_{\text{reconstruction loss } \mathcal{L}_0} + \underbrace{\sum_{t=1}^{T} \mathcal{L}_t}_{\text{diffusion loss}} \quad (10)$$

with

$$\mathcal{L}_t = D_{\text{KL}}\big(q(\boldsymbol{z}_{t-1}|\boldsymbol{z}_{\text{data}}, \boldsymbol{z}_t)||p_\theta(\boldsymbol{z}_{t-1}|\hat{\boldsymbol{z}}_{\text{data}}, \boldsymbol{z}_t)\big) \quad (11)$$

$$= \mathbb{E}_{\boldsymbol{\epsilon} \sim \mathcal{N}(\boldsymbol{0}, \boldsymbol{I})}\Big[\frac{1}{2}\Big(\frac{\text{SNR}(t-1)}{\text{SNR}(t)} - 1\Big)||\boldsymbol{\epsilon} - \hat{\boldsymbol{\epsilon}}_\theta||^2\Big] \quad (12)$$

during training. The prior loss should always be close to zero and can be computed exactly in closed form while the reconstruction loss must be estimated as described in Hoogeboom et al. (2022). In practice, however, we simply minimise the mean squared error $\mathcal{L}_{\text{train}} = \frac{1}{2}||\boldsymbol{\epsilon} - \hat{\boldsymbol{\epsilon}}||^2$ while randomly sampling time steps $t \sim \mathcal{U}(0, \dots, T)$, which is equivalent up to a multiplicative factor.

## B  NOTE ON EQUIVARIANCE OF THE CONDITIONAL MODEL

The 3D-conditional model can achieve equivariance without the usual "subspace-trick". The coordinates of pocket nodes provide a reference frame for all samples that can be used to translate them to a unique location (e.g. such that the pocket is centered at the origin: $\sum_i \boldsymbol{x}_i^{(P)} = \boldsymbol{0}$). By doing this for all training data, translation equivariance becomes irrelevant and the CoM-free subspace approach obsolete. To evaluate the likelihood of translated samples at inference time, we can first subtract the pocket's center of mass from the whole system and compute the likelihood after this mapping. Similarly, for sampling molecules we can first generate a ligand in a CoM-free version of the pocket and move the whole system back to the original location of the pocket nodes to restore translation equivariance. As long as the mean of our Gaussian noise distribution $p(\boldsymbol{z}_t|\boldsymbol{z}_{\text{data}}^{(P)}) = \mathcal{N}(\boldsymbol{\mu}(\boldsymbol{z}_{\text{data}}^{(P)}), \sigma^2\boldsymbol{I})$ depends equivariantly on the pocket node coordinates $\boldsymbol{x}^{(P)}$, $O(3)$-equivariance is satisfied as well (Appendix E). Since this change did not seem to affect the performance of the conditional model in our experiments, we decided to keep sampling in the linear subspace to ensure that the implementation is as similar as possible to the joint model, for which the subspace approach is necessary.

## C  IMPLEMENTATION DETAILS

**Molecule size**  As part of a sample's overall likelihood, we compute the empirical joint distribution of ligand and pocket nodes $p(N_L, N_P)$ observed in the training set and smooth it with a Gaussian filter ($\sigma = 1$). In the conditional generation scenario, we derive the distribution $p(N_L|N_P)$ and use it for likelihood computations.

For sampling, we can either fix molecule sizes manually or sample the number of ligand nodes from the same distribution given the number of nodes in the target pocket:

$$N_L \sim p(N_L|N_P). \quad (13)$$

For the experiments discussed in the main text, we increase the mean size of sampled molecules by 15 atoms to approximately match the sizes of molecules found in the test set. This modification makes the reported QuickVina scores more comparable as the *in silico* docking score is highly correlated with the molecular size, which is demonstrated in Figure 6. With the correction, the average size of generated molecules is 26.5, 28.6, and 26 respectively for DiffSBDD-cond ($C_\alpha$), DiffSBDD-inpaint ($C_\alpha$) and DiffSBDD-cond (full atom). Test set molecules from the Binding MOAD data set are composed of 28 atoms on average.

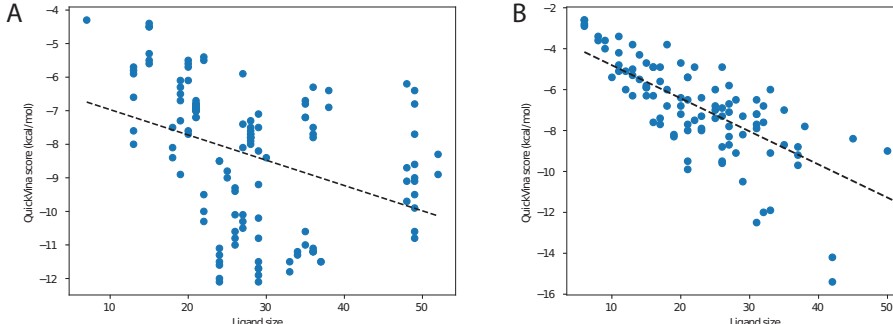

Figure 6: Correlation between ligand size and QuickVina score for reference molecules from the Binding MOAD (A) and CrossDocked (B) test sets.

**Preprocessing**  All molecules are expressed as graphs. For the $C_\alpha$ only model the node features for the protein are set as the one hot encoding of the amino acid type. The full atom model uses the same one hot encoding of atom types for ligand and protein nodes. We refrain from adding a categorical feature for distinguishing between protein and ligand atoms in this case and continue using two separate MLPs for embedding the node features instead.

**Noise schedule**  We use the pre-defined polynomial noise schedule introduced in (Hoogeboom et al., 2022):

$$\tilde{\alpha}_t = 1 - \left(\frac{t}{T}\right)^2, \quad t = 0, ..., T \tag{14}$$

Following (Nichol & Dhariwal, 2021; Hoogeboom et al., 2022), values of $\tilde{\alpha}_{t|s}^2 = \left(\frac{\tilde{\alpha}_t}{\tilde{\alpha}_s}\right)^2$ are clipped between 0.001 and 1 for numerical stability near $t = T$, and $\tilde{\alpha}_t$ is recomputed as

$$\tilde{\alpha}_t = \prod_{\tau=0}^{t} \tilde{\alpha}_{\tau|\tau-1}. \tag{15}$$

A tiny offset $\epsilon = 10^{-5}$ is used to avoid numerical problems at $t = 0$ defining the final noise schedule:

$$\alpha_t^2 = (1 - 2\epsilon) \cdot \tilde{\alpha}_t^2 + \epsilon. \tag{16}$$

**Feature scaling**  We scale the node type features $h$ by a factor of 0.25 relative to the coordinates $x$ which was empirically found to improve model perfomance in previous work (Hoogeboom et al., 2022).

**Hyperparameters**  Hyperparameters for all presented models are summarized in Table 3. Training takes about 2.5 h/3.75 h (cond/inpaint) for Binding MOAD in the $C_\alpha$ scenario and 12.5 h with full atom pocket representation on a single NVIDIA V100. For CrossDocked, 100 training epochs take approximately 24 h on an NVIDIA V100 GPU in the $C_\alpha$ case and 96 h per 100 epochs on a single NVIDIA A100 GPU with all atom pocket representation.

**Postprocessing**  For postprocessing of generated molecules, we use a similar procedure as in (Luo et al., 2021). Given a list of atom types and coordinates, bonds are first added using OpenBabel (O'Boyle et al., 2011). We then use RDKit to sanitise molecules, filter for the largest molecular fragment and finally remove steric clashes with 200 steps of force-field relaxation.

## C.1  EFFECT OF THE NUMBER OF RESAMPLING STEPS

In this section we discuss the number of resampling steps required for satisfactory results. To this end, we generated ligands for all test pockets with $r = 1$, $r = 5$, and $r = 10$ resampling steps, respectively. Because the resampling strategy slows down sampling approximately by a factor of $r$, we used the striding technique proposed by Nichol & Dhariwal (2021) and reduced the number of denoising steps proportionally to $r$. Nichol & Dhariwal (2021) showed that this approach reduces

Table 3: DiffSBDD hyperparameters.

|  | CrossDocked | | | Binding MOAD | | |
|---|---|---|---|---|---|---|
|  | Cond | Cond ($C_\alpha$) | Inpaint ($C_\alpha$) | Cond | Cond ($C_\alpha$) | Inpaint ($C_\alpha$) |
| No. layers | 6 | 6 | 6 | 5 | 5 | 5 |
| Joint embedding dim. | 32 | 32 | 32 | 32 | 32 | 32 |
| Hidden dim. | 256 | 256 | 256 | 128 | 128 | 128 |
| Learning rate | $10^{-4}$ | $10^{-4}$ | $10^{-4}$ | $5 \cdot 10^{-4}$ | $5 \cdot 10^{-4}$ | $5 \cdot 10^{-4}$ |
| Weight decay | $10^{-12}$ | $10^{-12}$ | $10^{-12}$ | $10^{-12}$ | $10^{-12}$ | $10^{-12}$ |
| Diffusion steps | 1000 | 1000 | 1000 | 500 | 500 | 500 |
| Edges (ligand-ligand) | $< 7\,\text{Å}$ | fully connected | fully connected | fully connected | fully connected | fully connected |
| Edges (ligand-pocket) | $< 7\,\text{Å}$ | fully connected | fully connected | $< 5\,\text{Å}$ | $< 8\,\text{Å}$ | $< 8\,\text{Å}$ |
| Edges (pocket-pocket) | $< 7\,\text{Å}$ | fully connected | fully connected | $< 5\,\text{Å}$ | $< 8\,\text{Å}$ | $< 8\,\text{Å}$ |
| Epochs | 1000 | 1000 | 1000 | 800 | 800 | 800 |

Table 4: Evaluation of generated molecules for target pockets from the Binding MOAD test set with the inpainting approach and $C_\alpha$ pocket representation for varying numbers of resampling steps $r$ and denoising steps $T$.

| $r$ | $T$ | Vina Score (kcal/mol, $\downarrow$) | QED ($\uparrow$) | SA ($\uparrow$) | Lipinski ($\uparrow$) | Diversity ($\uparrow$) | Time (s, $\downarrow$) |
|---|---|---|---|---|---|---|---|
| 1 | 500 | $-5.601 \pm 1.92$ | $0.495 \pm 0.12$ | $0.358 \pm 0.09$ | $4.910 \pm 0.31$ | $0.850 \pm 0.04$ | $40.298 \pm 13.52$ |
| 5 | 100 | $-5.963 \pm 1.93$ | $0.541 \pm 0.13$ | $0.365 \pm 0.09$ | $4.946 \pm 0.24$ | $0.853 \pm 0.05$ | $45.074 \pm 21.14$ |
| 10 | 50 | $-6.080 \pm 2.14$ | $0.554 \pm 0.13$ | $0.367 \pm 0.09$ | $4.957 \pm 0.21$ | $0.855 \pm 0.05$ | $41.490 \pm 14.32$ |

the number of sampling steps significantly without sacrificing sample quality. In our case, it allows us to retain sampling speed while increasing the number of resampling steps.

To gauge the effect of resampling for molecule generation we show the distribution of RMSD values between the center of mass of reference molecules and generated molecules in Figure 7. The unmodified replacement method ($r = 1$) produces molecules that are clearly farther away from the presumed pocket center than the conditional model. Increasing $r$ moves the mean distance closer to the average displacement of molecules from the conditional method. This effect seems to saturate at $r = 10$ which is in line with the results obtained for images (Lugmayr et al., 2022).

Table 4 shows that neither the additional resampling steps nor the shortened denoising trajectory degrade the performance on the reported molecular metrics. The average docking scores even improve slightly which might reflect better positioning of generated ligands in the pockets prior to docking. The same model trained with $T = 500$ diffusion steps was used in all three cases. These experiments have been conducted without the adjustment of molecule sizes described in Section C.

## D BINDING MOAD DATASET

We curate a dataset of experimentally determined complexed protein-ligand structures from Binding MOAD (Hu et al., 2005). We keep pockets with valid[5] and moderately 'drug-like' ligands with QED score $> 0.3$. We further discard small molecules that contain atom types $\notin \{C, N, O, S, B, Br, Cl, P, I, F\}$ as well as binding pockets with non-standard amino acids. We define binding pockets as the set of residues that have any atom within $8\,\text{Å}$ of any ligand atom. Ligand redundancy is reduced by randomly sampling at most 50 molecules with the same chemical component identifier (3-letter-code). After removing corrupted entries that could not be processed, $40\,354$ training pairs and 130 testing pairs remain. A validation set of size 246 is used to monitor estimated log-likelihoods during training. The split is made to ensure different sets do not contain proteins from the same Enzyme Commission Number (EC Number) main class.

## E PROOFS

In the following proofs we do not consider categorical node features $\boldsymbol{h}$ as only the positions $\boldsymbol{x}$ are subject to equivariance constraints. Furthermore, we do not distinguish between the zeroth latent

---

[5]as defined in `http://www.bindingmoad.org/`

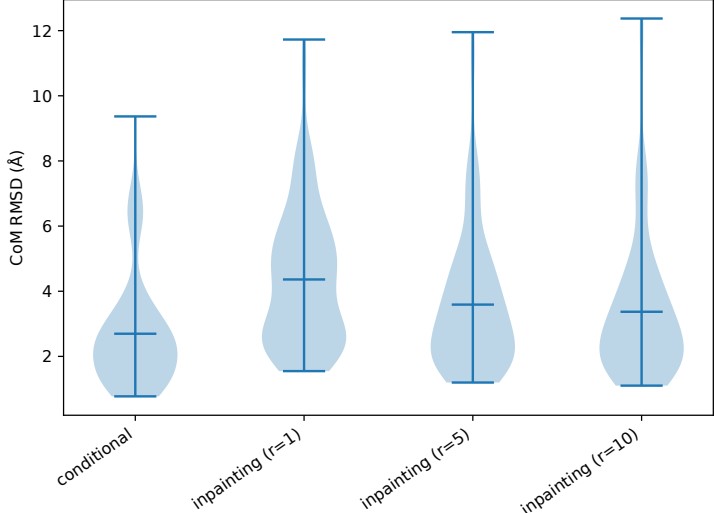

Figure 7: RMSD between reference molecules' center of mass and generated molecules' center of mass for the conditional model and inpaining model with varying numbers of resampling steps $r$. The pocket representation is $C_\alpha$ in all cases.

representation $\boldsymbol{x}_0$ and data domain representations $\boldsymbol{x}_{\text{data}}$ for ease of notation, and simply drop the subscripts.

## E.1   $O(3)$-EQUIVARIANCE OF THE PRIOR PROBABILITY

The isotropic Gaussian prior $p(\boldsymbol{x}_T^{(L)}|\boldsymbol{x}^{(P)}) = \mathcal{N}(\boldsymbol{\mu}(\boldsymbol{x}^{(P)}), \sigma^2\boldsymbol{I})$ is equivariant to rotations and reflections represented by an orthogonal matrix $\boldsymbol{R} \in \mathbb{R}^{3\times3}$ as long as $\boldsymbol{\mu}(\boldsymbol{R}\boldsymbol{x}^{(P)}) = \boldsymbol{R}\boldsymbol{\mu}(\boldsymbol{x}^{(P)})$ because:

$$
\begin{aligned}
p(\boldsymbol{R}\boldsymbol{x}_T^{(L)}|\boldsymbol{R}\boldsymbol{x}^{(P)}) &= \frac{1}{\sqrt{(2\pi)^{N_L}\sigma^2}} \exp\left(-\frac{1}{2\sigma^2}||\boldsymbol{R}\boldsymbol{x}_T^{(L)} - \boldsymbol{\mu}(\boldsymbol{R}\boldsymbol{x}^{(P)})||^2\right) \\
&= \frac{1}{\sqrt{(2\pi)^{N_L}\sigma^2}} \exp\left(-\frac{1}{2\sigma^2}||\boldsymbol{R}\boldsymbol{x}_T^{(L)} - \boldsymbol{R}\boldsymbol{\mu}(\boldsymbol{x}^{(P)})||^2\right) \\
&= \frac{1}{\sqrt{(2\pi)^{N_L}\sigma^2}} \exp\left(-\frac{1}{2\sigma^2}||\boldsymbol{R}(\boldsymbol{x}_T^{(L)} - \boldsymbol{\mu}(\boldsymbol{x}^{(P)}))||^2\right) \\
&= \frac{1}{\sqrt{(2\pi)^{N_L}\sigma^2}} \exp\left(-\frac{1}{2\sigma^2}||\boldsymbol{x}_T^{(L)} - \boldsymbol{\mu}(\boldsymbol{x}^{(P)})||^2\right) \\
&= p(\boldsymbol{x}_T^{(L)}|\boldsymbol{x}^{(P)}).
\end{aligned}
$$

Here we used $||\boldsymbol{R}\boldsymbol{x}||_2 = ||\boldsymbol{x}||_2$ for orthogonal $\boldsymbol{R}$.

## E.2   $O(3)$-EQUIVARIANCE OF THE TRANSITION PROBABILITIES

The denoising transition probabilities from time step $t$ to $s < t$ are defined as isotropic normal distributions:

$$
p_\theta(\boldsymbol{x}_{t-1}^{(L)}|\boldsymbol{x}_t^{(L)}, \hat{\boldsymbol{x}}^{(L)}, \boldsymbol{x}^{(P)}) = \mathcal{N}(\boldsymbol{x}_{t-1}^{(L)}|\boldsymbol{\mu}_{t\to s}(\boldsymbol{x}_t^{(L)}, \hat{\boldsymbol{x}}^{(L)}, \boldsymbol{x}^{(P)}), \sigma_{t\to s}^2\boldsymbol{I}). \tag{17}
$$

Therefore, $p_\theta(\boldsymbol{x}_{t-1}^{(L)}|\boldsymbol{x}_t^{(L)}, \hat{\boldsymbol{x}}^{(L)}, \boldsymbol{x}^{(P)})$ is $O(3)$-equivariant by a similar argument to Section E.1 if $\boldsymbol{\mu}_{t\to s}$ is computed equivariantly from the three-dimensional context.

Recalling the definition of $\boldsymbol{\mu}_{t\to s} = \frac{\alpha_{t|s}\sigma_s^2}{\sigma_t^2}\boldsymbol{x}_t^{(L)} + \frac{\alpha_s\sigma_{t|s}^2}{\sigma_t^2}\hat{\boldsymbol{x}}^{(L)}$, we can prove its equivariance as follows:

$$\boldsymbol{\mu}_{t\to s}(\boldsymbol{R}\boldsymbol{x}_t^{(L)}, \boldsymbol{R}\boldsymbol{x}^{(P)}) = \frac{\alpha_{t|s}\sigma_s^2}{\sigma_t^2}\boldsymbol{R}\boldsymbol{x}_t^{(L)} + \frac{\alpha_s\sigma_{t|s}^2}{\sigma_t^2}\hat{\boldsymbol{x}}^{(L)}(\boldsymbol{R}\boldsymbol{x}_t^{(L)}, \boldsymbol{R}\boldsymbol{x}^{(P)})$$

$$= \frac{\alpha_{t|s}\sigma_s^2}{\sigma_t^2}\boldsymbol{R}\boldsymbol{x}_t^{(L)} + \frac{\alpha_s\sigma_{t|s}^2}{\sigma_t^2}\boldsymbol{R}\hat{\boldsymbol{x}}^{(L)}(\boldsymbol{x}_t^{(L)}, \boldsymbol{x}^{(P)}) \quad \text{(equivariance of } \hat{\boldsymbol{x}}^{(L)})$$

$$= \boldsymbol{R}\Big(\frac{\alpha_{t|s}\sigma_s^2}{\sigma_t^2}\boldsymbol{x}_t^{(L)} + \frac{\alpha_s\sigma_{t|s}^2}{\sigma_t^2}\hat{\boldsymbol{x}}^{(L)}(\boldsymbol{x}_t^{(L)}, \boldsymbol{x}^{(P)})\Big)$$

$$= \boldsymbol{R}\boldsymbol{\mu}_{t\to s}(\boldsymbol{x}_t^{(L)}, \boldsymbol{x}^{(P)}),$$

where $\hat{\boldsymbol{x}}^{(L)}$ defined as $\hat{\boldsymbol{x}}^{(L)} = \frac{1}{\alpha_t}\boldsymbol{x}_t^{(L)} - \frac{\sigma_t}{\alpha_t}\hat{\boldsymbol{\epsilon}}$ is equivariant because:

$$\hat{\boldsymbol{x}}^{(L)}(\boldsymbol{R}\boldsymbol{x}_t^{(L)}, \boldsymbol{R}\boldsymbol{x}^{(P)}) = \frac{1}{\alpha_t}\boldsymbol{R}\boldsymbol{x}_t^{(L)} - \frac{\sigma_t}{\alpha_t}\hat{\boldsymbol{\epsilon}}(\boldsymbol{R}\boldsymbol{x}_t^{(L)}, \boldsymbol{R}\boldsymbol{x}^{(P)}, t)$$

$$= \frac{1}{\alpha_t}\boldsymbol{R}\boldsymbol{x}_t^{(L)} - \frac{\sigma_t}{\alpha_t}\boldsymbol{R}\hat{\boldsymbol{\epsilon}}(\boldsymbol{x}_t^{(L)}, \boldsymbol{x}^{(P)}, t) \quad (\hat{\boldsymbol{\epsilon}} \text{ predicted by equivariant neural network)}$$

$$= \boldsymbol{R}\Big(\frac{1}{\alpha_t}\boldsymbol{x}_t^{(L)} - \frac{\sigma_t}{\alpha_t}\hat{\boldsymbol{\epsilon}}(\boldsymbol{x}_t^{(L)}, \boldsymbol{x}^{(P)}, t)\Big)$$

$$= \boldsymbol{R}\hat{\boldsymbol{x}}^{(L)}(\boldsymbol{x}_t^{(L)}, \boldsymbol{x}^{(P)}).$$

### E.3 $O(3)$-EQUIVARIANCE OF THE LEARNED LIKELIHOOD

Let $\boldsymbol{R} \in \mathbb{R}^{3\times3}$ be an orthogonal matrix representing an element $g$ from the general orthogonal group $O(3)$. We obtain the marginal probability density of the Markovian denoising process as follows

$$p_\theta(\boldsymbol{x}_0^{(L)}|\boldsymbol{x}^{(P)}) = \int p(\boldsymbol{x}_T^{(L)}|\boldsymbol{x}^{(P)})p_\theta(\boldsymbol{x}_{0:T-1}^{(L)}|\boldsymbol{x}_T^{(L)}, \boldsymbol{x}^{(P)})\mathrm{d}\boldsymbol{x}_{1:T}$$

$$= \int p(\boldsymbol{x}_T^{(L)}|\boldsymbol{x}^{(P)})\prod_{t=1}^T p_\theta(\boldsymbol{x}_{t-1}^{(L)}|\boldsymbol{x}_t^{(L)}, \boldsymbol{x}^{(P)})\mathrm{d}\boldsymbol{x}_{1:T}$$

and the sample's likelihood is $O(3)$-equivariant:

$$p_\theta(\boldsymbol{R}\boldsymbol{x}_0^{(L)}|\boldsymbol{R}\boldsymbol{x}^{(P)}) = \int p(\boldsymbol{R}\boldsymbol{x}_T^{(L)}|\boldsymbol{R}\boldsymbol{x}^{(P)})\prod_{t=1}^T p_\theta(\boldsymbol{R}\boldsymbol{x}_{t-1}^{(L)}|\boldsymbol{R}\boldsymbol{x}_t^{(L)}, \boldsymbol{R}\boldsymbol{x}^{(P)})\mathrm{d}\boldsymbol{x}_{1:T}$$

$$= \int p(\boldsymbol{x}_T^{(L)}|\boldsymbol{x}^{(P)})\prod_{t=1}^T p_\theta(\boldsymbol{R}\boldsymbol{x}_{t-1}^{(L)}|\boldsymbol{R}\boldsymbol{x}_t^{(L)}, \boldsymbol{R}\boldsymbol{x}^{(P)})\mathrm{d}\boldsymbol{x}_{1:T} \quad \text{(equivariant prior)}$$

$$= \int p(\boldsymbol{x}_T^{(L)}|\boldsymbol{x}^{(P)})\prod_{t=1}^T p_\theta(\boldsymbol{x}_{t-1}^{(L)}|\boldsymbol{x}_t^{(L)}, \boldsymbol{x}^{(P)})\mathrm{d}\boldsymbol{x}_{1:T} \quad \text{(equivariant transition probabilities)}$$

$$= p_\theta(\boldsymbol{x}_0^{(L)}|\boldsymbol{x}^{(P)})$$

## F  $SE(3)$-EQUIVARIANT GRAPH NEURAL NETWORK

Chiral molecules cannot be superimposed by combination of rotations and translations. Instead they are mirrored along a stereocenter, axis, or plane. As chirality can significantly alter a molecule's chemical properties, we use a variant of the $E(3)$-equivariant graph neural networks (Satorras et al., 2021) presented in Equations (3)-(5) that is sensitive to reflections and hence $SE(3)$-equivariant. We change the coordinate update equation, Equ. (5), of standard EGNNs in the following way

$$\boldsymbol{x}_i^{l+1} = \boldsymbol{x}_i^l + \sum_{j\neq i}\frac{\boldsymbol{x}_i^l - \boldsymbol{x}_j^l}{d_{ij}+1}\phi_x^d(\boldsymbol{h}_i^l, \boldsymbol{h}_j^l, d_{ij}^2, a_{ij}) + \frac{(\boldsymbol{x}_i^l - \bar{\boldsymbol{x}}^l)\times(\boldsymbol{x}_j^l - \bar{\boldsymbol{x}}^l)}{||(\boldsymbol{x}_i^l - \bar{\boldsymbol{x}}^l)\times(\boldsymbol{x}_j^l - \bar{\boldsymbol{x}}^l)||+1}\phi_x^\times(\boldsymbol{h}_i^l, \boldsymbol{h}_j^l, d_{ij}^2, a_{ij}),$$

$$(18)$$

where $\bar{\boldsymbol{x}}^l$ denotes the center of mass of all nodes at layer $l$. This modification makes the EGNN layer sensitive to reflections while staying close to the original formalism. Since the resulting graph neural networks are only equivariant to the $SE(3)$ group, we will hereafter call them SEGNNs for short.

## F.1 DISCUSSION OF EQUIVARIANCE

Here we study how the suggested change in the coordinate update equation breaks reflection symmetry while preserving equivariance to rotations. Messages and scalar feature updates (Equations (3) and (4)) remain $E(3)$-invariant as in the original model and are therefore not considered in this section. We analyze transformations composed of a translation by $\boldsymbol{t} \in \mathbb{R}^3$ and a rotation/reflection by an orthogonal matrix $\boldsymbol{R} \in \mathbb{R}^{3\times3}$ with $\boldsymbol{R}^T\boldsymbol{R} = \boldsymbol{I}$. The output at layer $l+1$ given the transformed input $\boldsymbol{R}\boldsymbol{x}_i^l + \boldsymbol{t}$ at layer $l$ is calculated as:

$$\boldsymbol{R}\boldsymbol{x}_i^l + \boldsymbol{t} + \sum_{j\neq i} \frac{\boldsymbol{R}\boldsymbol{x}_i^l + \boldsymbol{t} - (\boldsymbol{R}\boldsymbol{x}_j^l + \boldsymbol{t})}{d_{ij} + 1} \phi_x^d(\cdot) + \frac{(\boldsymbol{R}\boldsymbol{x}_i^l + \boldsymbol{t} - (\boldsymbol{R}\bar{\boldsymbol{x}}^l + \boldsymbol{t})) \times (\boldsymbol{R}\boldsymbol{x}_j^l + \boldsymbol{t} - (\boldsymbol{R}\bar{\boldsymbol{x}}^l + \boldsymbol{t}))}{Z_{ij}^\times + 1} \phi_x^\times(\cdot)$$

(19)

$$= \boldsymbol{R}\boldsymbol{x}_i^l + \boldsymbol{t} + \sum_{j\neq i} \frac{\boldsymbol{R}(\boldsymbol{x}_i^l - \boldsymbol{x}_j^l)}{d_{ij} + 1} \phi_x^d(\cdot) + \frac{(\boldsymbol{R}\boldsymbol{x}_i^l - \boldsymbol{R}\bar{\boldsymbol{x}}^l) \times (\boldsymbol{R}\boldsymbol{x}_j^l - \boldsymbol{R}\bar{\boldsymbol{x}}^l)}{Z_{ij}^\times + 1} \phi_x^\times(\cdot)$$

(20)

$$= \boldsymbol{R}\boldsymbol{x}_i^l + \boldsymbol{t} + \sum_{j\neq i} \frac{\boldsymbol{R}(\boldsymbol{x}_i^l - \boldsymbol{x}_j^l)}{d_{ij} + 1} \phi_x^d(\cdot) + \frac{\det(\boldsymbol{R})\boldsymbol{R}\big((\boldsymbol{x}_i^l - \bar{\boldsymbol{x}}^l) \times (\boldsymbol{x}_j^l - \bar{\boldsymbol{x}}^l)\big)}{Z_{ij}^\times + 1} \phi_x^\times(\cdot)$$

(21)

$$= \boldsymbol{R}\boldsymbol{x}_i^{l+1} + \boldsymbol{t} + \big(\det(\boldsymbol{R}) - 1\big) \sum_{j\neq i} \frac{\boldsymbol{R}\big((\boldsymbol{x}_i^l - \bar{\boldsymbol{x}}^l) \times (\boldsymbol{x}_j^l - \bar{\boldsymbol{x}}^l)\big)}{Z_{ij}^\times + 1} \cdot$$

(22)

This result shows that the output coordinates are only equivariantly transformed if $\boldsymbol{R}$ is orientation preserving, i.e. $\det(\boldsymbol{R}) = 1$. If $\boldsymbol{R}$ is a reflection ($\det(\boldsymbol{R}) = -1$), coordinates will be updated with an additional summand that breaks the symmetry.

The learnable coefficients $\phi_x^d(\boldsymbol{h}_i^l, \boldsymbol{h}_j^l, d_{ij}^2, a_{ij})$ and $\phi_x^\times(\boldsymbol{h}_i^l, \boldsymbol{h}_j^l, d_{ij}^2, a_{ij})$ only depend on relative distances and are therefore $E(3)$-invariant. Their arguments are represented with the "·" symbol for brevity. Likewise, the normalization factor $||(\boldsymbol{x}_i^l - \bar{\boldsymbol{x}}^l) \times (\boldsymbol{x}_j^l - \bar{\boldsymbol{x}}^l)||$ is abbreviated as $Z_{ij}^\times$. Already in the first line we used the fact that the mean transforms equivariantly. Furthermore, we use $\boldsymbol{R}\boldsymbol{a} \times \boldsymbol{R}\boldsymbol{b} = \det(\boldsymbol{R})\boldsymbol{R}(\boldsymbol{a} \times \boldsymbol{b})$ in the second step, which can be derived as follows:

$$\boldsymbol{x}^T(\boldsymbol{R}\boldsymbol{a} \times \boldsymbol{R}\boldsymbol{b}) = \det(\underbrace{[\boldsymbol{x}, \boldsymbol{R}\boldsymbol{a}, \boldsymbol{R}\boldsymbol{b}]}_{\in \mathbb{R}^{3\times3}})$$

(23)

$$= \det(\boldsymbol{R}[\boldsymbol{R}^T\boldsymbol{x}, \boldsymbol{a}, \boldsymbol{b}])$$

(24)

$$= \det(\boldsymbol{R})\det([\boldsymbol{R}^T\boldsymbol{x}, \boldsymbol{a}, \boldsymbol{b}])$$

(25)

$$= \det(\boldsymbol{R})\big(\boldsymbol{x}^T\boldsymbol{R}(\boldsymbol{a} \times \boldsymbol{b})\big)$$

(26)

$$= \boldsymbol{x}^T\big(\det(\boldsymbol{R})\boldsymbol{R}(\boldsymbol{a} \times \boldsymbol{b})\big)$$

(27)

The stated property of the cross product follows because this derivation is true for all $\boldsymbol{x} \in \mathbb{R}^3$.

## F.2 Empirical Results

To show the effectiveness of this architecture on a simple toy example, we repeat the classification experiment by Adams et al. (2021) who train neural networks to classify tetrahedral chiral centers as right-handed (*rectus*, 'R') or left-handed (*sinister*, 'S'). We closely follow their data split and experimental set-up and only replace the classifier with EGNN and SEGNNs, respectively. The results in Table 5 clearly demonstrate that the $SE(3)$-equivariant EGNN is capable of solving this task (without any hyperparameter optimization) whereas the $E(3)$-equivariant version does not do better than random guessing.

Table 5: Accuracy on the R/S classification task. Results in the first section are taken from (Adams et al., 2021) and included for reference.

| Model | R/S Accuracy (%) |
|---|---|
| ChIRo | 98.5 |
| SchNet | 54.4 |
| DimeNet++ | 65.7 |
| SphereNet | 98.2 |
| EGNN | 50.4 |
| SEGNN | 83.4 |

# G    Extended results

## G.1 Additional Experimental Details

The numbers of available molecules differ slightly between different methods due to computational issues or missing molecules in the available baseline sets. More precisely, on average 93.5, 92.8, and 98.3 molecules have been evaluated per pocket for DiffSBDD-cond, DiffSBDD-inpaint ($C_\alpha$), and DiffSBDD-cond ($C_\alpha$), respectively. For Pocket2Mol, 98.4 molecules are available per pocket. The set of 3D-SBDD molecules does not contain generated ligands for two test pockets. For the remaining 98 pockets, 89.9 molecules are available on average.

All Figures show molecules generated where the starting number of nodes equals the number of nodes in the reference ligands, with the exception of Figure 4, which employs the sampling strategy outlined in Appendix C.

## G.2 Additional Molecular Metrics

In addition to the molecular properties discussed in Section 4 we assess the models' ability to produce novel and valid molecules using four simple metrics: validity, connectivity, uniqueness, and novelty. **Validity** measures the proportion of generated molecules that pass basic tests by RDKit–mostly ensuring correct valencies. **Connectivity** is the proportion of valid molecules that do not contain any disconnected fragments. We convert every valid and connected molecule from a graph into a canonical SMILES string representation, count the number unique occurrences in the set of generated molecules and compare those to the training set SMILES to compute **uniqueness** and **novelty** respectively.

Table 6 shows that only a small fraction of all generated molecules is invalid and must be discarded for downstream processing. The DiffSBDD models trained on CrossDocked with $C_\alpha$ pocket representation generate fragmented molecules about 50% of the time. Since we can simply select and process the largest fragments in these cases, low connectivity does not necessarily affect the efficiency of the generative process. Moreover, all models produce diverse sets of molecules unseen in the training set.

## G.3 Octanol-water partition coefficient

The octanol-water partition coefficient ($\log P$) is a measure of lipophilicity and is commonly reported for potential drug candidates (Wildman & Crippen, 1999). We summarize this property for our generated molecules in Table 7.

Table 6: Basic molecular metrics for generated small molecules given a $C_\alpha$ and full atom representation of the protein pocket.

| Model | Validity | Connectivity | Uniqueness | Novelty |
|---|---|---|---|---|
| CrossDocked Training data | 100% | 100% | – | – |
| DiffSBDD-cond ($C_\alpha$) | 97.75% | 48.02% | 96.95% | 100% |
| DiffSBDD-inpaint ($C_\alpha$) | 91.62% | 51.38% | 98.64% | 100% |
| DiffSBDD-cond | 93.23% | 83.46% | 97.46% | 100% |
| Binding MOAD Training data | 96.38% | 100% | – | – |
| DiffSBDD-cond ($C_\alpha$) | 92.51% | 52.13% | 100.00% | 100.00% |
| DiffSBDD-inpaint ($C_\alpha$) | 90.28% | 73.19% | 100.00% | 100.00% |
| DiffSBDD-cond | 95.39% | 39.58% | 100.00% | 100.00% |

Table 7: LogP values of generated molecules.

| | CrossDocked | Binding MOAD |
|---|---|---|
| Test set | $0.894 \pm 2.73$ | $0.456 \pm 1.15$ |
| 3D-SBDD (AR) (Luo et al., 2021) | $0.273 \pm 2.01$ | — |
| Pocket2Mol (Peng et al., 2022) | $1.720 \pm 1.97$ | — |
| DiffSBDD-cond ($C_\alpha$) | $-0.184 \pm 1.01$ | $0.110 \pm 1.03$ |
| DiffSBDD-inpaint ($C_\alpha$) | $-0.519 \pm 1.09$ | $0.574 \pm 1.39$ |
| DiffSBDD-cond | $-0.328 \pm 1.18$ | $0.845 \pm 1.61$ |

## G.4 OPTIMIZATION

We demonstrate the effect the number of noising/denoising steps ($t$) has on various molecular properties in Figure 8. We test all values of $t$ at intervals of 10 steps and 200 molecules are sampled at every timestep. Note this does not allow for explicit optimization of any particular property unless combined with the evolutionary algorithm.

During the evolutionary algorithm, at the end of every generation the top 10 docking molecules are used to seed the next population. Every seed molecule is elaborated into 20 new candidates with a randomly chosen $t$ between 10 and 150. To make the first population, we start with the single reference molecule and sample 200 new molecules with $t$ chosen as above.

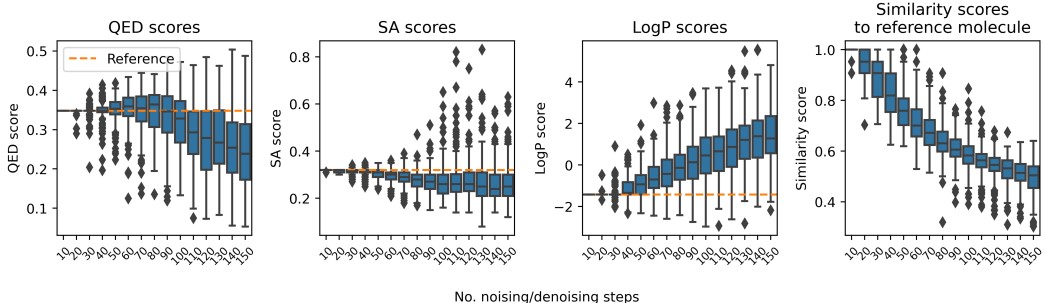

Figure 8: Effect of number of noising/denoising steps on molecule properties.

## G.5 AGREEMENT OF GENERATED AND DOCKED CONFORMATIONS

Here we discuss an alternative way of using QuickVina for assessing the quality of the conditional generation procedure besides its *in silico* docking score. We compare the generated raw conformations (before force-field relaxation) to the best scoring QuickVina docking pose and plot the

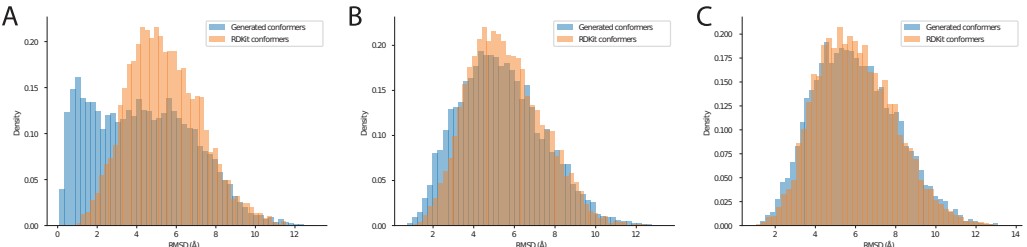

Figure 9: RMSD between original and docked conformations for the CrossDocked dataset. (A) DiffSBDD-cond, sample size 8804. (B) DiffSBDD-cond ($C_\alpha$), sample size 9611. (C) DiffSBDD-inpaint ($C_\alpha$), sample size 8641.

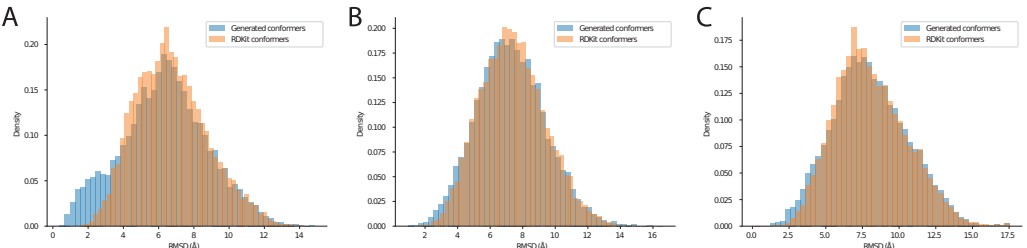

Figure 10: RMSD between original and docked conformations for the Binding MOAD dataset. (A) DiffSBDD-cond, sample size 12 520. (B) DiffSBDD-cond ($C_\alpha$), sample size 12 641. (C) DiffSBDD-inpaint ($C_\alpha$), sample size 12 902.

distribution of resulting RMSD values in Figures 9 and 10. As a baseline, the procedure is repeated for RDKit conformers of the same molecules with identical center of mass. For a large percentage of molecules generated by the all-atom CrossDocked model, QuickVina agrees with the predicted bound conformations, leaving them almost unchanged (RMSD below 2 Å). This demonstrates successful conditioning on the given protein pockets.

For the $C_\alpha$-only models results are less convincing. They produce poses that only slightly improve upon conformers lacking pocket-context. Likely, this is caused by atomic clashes with the proteins' side chains that QuickVina needs to resolve.

### G.6 RANDOM GENERATED MOLECULES

Randomly selected molecules generated with our method and 3 baseline methods (LiGAN, SBDD-3D and Pocket2Mol) when trained with CrossDocked are presented in Figure 11. Randomly selected molecules generated by our method when trained with Binding MOAD are show in Figure 12.

### G.7 DISTRIBUTION OF DOCKING SCORES BY TARGET

We present extensive evaluation of the docking scores for our generated molecules in Figure 13. We evaluate all models trained with a given dataset first against all targets (Figure 13A+C) and 10 randomly chosen targets (Figure 13B+D). We note that the all-atom model trained using CrossDocked data outperforms all other methods. Unsurprisingly, model performance is highly target dependent, likely varying with properties like pocket geometry, size, charge, and hydrophbicity, which would affect the propensity of generating high affinity molecules.

## H RELATED WORK

**Diffusion Models for Molecules** Inspired by non-equilibrium thermodynamics, diffusion models have been proposed to learn data distributions by modeling a denoising (reverse diffusion) process and have achieved remarkable success in a variety of tasks such as image, audio synthesis and point

| Target | LiGAN | | | SBDD-3D (AR) | | | pocket2mol | | | DiffSBDD-cond | | |
|--------|-------|--|--|--------------|--|--|------------|--|--|---------------|--|--|
| 2jjg | | | | | | | | | | | | |
| 3pnm | | | | | | | | | | | | |
| 1afs | | | | | | | | | | | | |
| 14gs | | | | | | | | | | | | |
| 4tos | | | | | | | | | | | | |
| 3li4 | | | | | | | | | | | | |
| 4yhj | | | | | | | | | | | | |
| 3pnm | | | | | | | | | | | | |
| 3kc1 | | | | | | | | | | | | |
| 2pc8 | | | | | | | | | | | | |

Figure 11: Generated molecules for 10 randomly chosen targets in the CrossDocked test set. For each target, 3 randomly selected generated molecules from 4 models are shown.

| Target | DiffSBDD-cond | | | | | DiffSBDD-inpaint | | | | |
|--------|---|---|---|---|---|---|---|---|---|---|
| 2fky | | | | | | | | | | |
| 3zjx | | | | | | | | | | |
| 3gt9 | | | | | | | | | | |
| 5ndu | | | | | | | | | | |
| 2vl8 | | | | | | | | | | |
| 1j78 | | | | | | | | | | |
| 3eks | | | | | | | | | | |
| 5zzb | | | | | | | | | | |
| 1fd7 | | | | | | | | | | |
| 2a5x | | | | | | | | | | |

Figure 12: Generated molecules for 10 randomly chosen targets in the Binding MOAD test set. For each target-model pair, 5 randomly selected generated molecules are shown. $C_\alpha$ level proteins were used for both models.

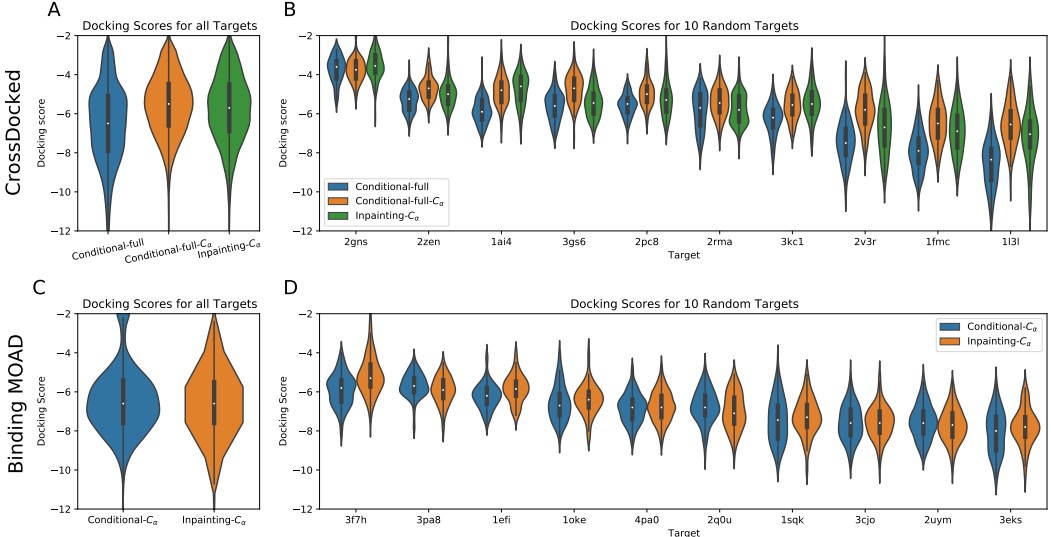

Figure 13: Docking scores of generated molecules for various methods trained on the CrossDocked (A-B) and Binding MOAD (C-D) datasets. (A) Violin plot of docking scores for all 3 methods trained using CrossDocked. (B) Same as before but for 10 randomly chosen targets sorted by mean score. (C) Violin plot of docking scores for all 2 methods trained using Binding MOAD. (D) Same as before but for 10 randomly chosen targets sorted by mean score.

cloud generation (Kingma et al., 2021; Kong et al., 2021; Luo & Hu, 2021). Recently, efforts have been made to utilize diffusion models for molecule design (Du et al., 2022b). Specifically, Hoogeboom et al. (2022) propose a diffusion model with an equivariant network that operates both on continuous atomic coordinates and categorical atom types to generate new molecules in 3D space. Torsional Diffusion (Jing et al., 2022) focuses on a conditional setting where molecular conformations (atomic coordinates) are generated from molecular graphs (atom types and bonds). Similarly, 3D diffusion models have been applied to generative design of larger biomolecular structures, such as antibodies (Luo et al., 2022) and other proteins (Anand & Achim, 2022; Trippe et al., 2022).

**Structure-based Drug Design** Structure-based Drug Design (SBDD) (Ferreira et al., 2015; Anderson, 2003) relies on the knowledge of the 3D structure of the biological target obtained either through experimental methods or high-confidence predictions using homology modelling (Kelley et al., 2015). Candidate molecules are then designed to bind with high affinity and specificity to the target using interactive software (Kalyaanamoorthy & Chen, 2011) and often human-based intuition (Ferreira et al., 2015). Recent advances in deep generative models have brought a new wave of research that model the conditional distribution of ligands given biological targets and thus enable *de novo* structure-based drug design. Most of recent work consider this task as a sequential generation problem and design a variety of generative methods including autoregressive models, reinforcement learning, etc., to generate ligands inside protein pockets atom by atom (Drotár et al., 2021; Luo et al., 2021; Li et al., 2021; Peng et al., 2022).

**Geometric Deep Learning for Drug Discovery** Geometric deep learning refers to incorporating geometric priors in neural architecture design that respects symmetry and invariance, thus reduces sample complexity and eliminates the need for data augmentation (Bronstein et al., 2021). It has been prevailing in a variety of drug discovery tasks from virtual screening to de novo drug design as symmetry widely exists in the representation of drugs. One line of work introduces graph and geometry priors and designs message passing neural networks and equivariant neural networks that are permutation- and translation-, rotation-, reflection-equivariant, respectively (Duvenaud et al., 2015; Gilmer et al., 2017; Satorras et al., 2021; Lapchevskyi et al., 2020; Du et al., 2022a), and has been widely used in representing biomolecules from small molecules to proteins (Atz et al., 2021) and solving downstream tasks such as molecular property prediction (Schütt et al., 2018; Klicpera et al., 2020), binding pose prediction (Stärk et al., 2022) or molecular dynamics (Batzner et al., 2022;

Holdijk et al., 2022). Another line of work focuses on generative design of new molecules (Du et al., 2022b;c). Specifically, they formulate molecule design as a graph or geometry generation problem and there are two strategies: one-shot generation that generates graphs (atom and bond features) in one step and sequential generation that generates them in a sequence of steps.

