# OpenReview forum: "Structure-based Drug Design with Equivariant Diffusion Models"
_ICLR.cc/2023/Conference — Submitted to ICLR 2023_

### Official Review · Reviewer_bBKg · 2022-10-21

**Confidence:** 4
**Correctness:** 3
**Technical Novelty And Significance:** 3
**Empirical Novelty And Significance:** 2
**Recommendation:** 3

**Clarity, Quality, Novelty And Reproducibility:**

Clarity is pretty good with all technical and experimental details greatly explained.

Quality is high but can still be improved. E.g., There is an anonymous reference above Equation.10. However, even the title of the anonymous reference is missed, making the sentence a little confusing...

Novelty is good by also limited. See details in the "weakness" section.

Reproducibility is supposed to be good since the source code is nicely provided, though I didn't take a closer look at the details of the provided code.

**Strength And Weaknesses:**

Strength:
1. The paper is clearly written. Both the task and the method are clear and easy to follow.
2. For the specific task of "structure-based drug design", the method is the first diffusion-based model implemented.
3. The author nicely collected a new dataset for benchmarking this task.

Weakness:
1. Though the first diffusion for this specific task, the method itself of equivariant diffusion is not new. Besides, the task has also been well formulated in previous ML papers. Thus the papers seem to be a little incremental for ML conferences, with limited technical contributions.
2. For the proposed method, the author also discussed the "inpainting" scenario. However, it's very unclear to me how this could be useful? It feels like a direct and unnatural adoption of diffusion models on images, but in my mind perturbing the protein targets is not very meaningful in the biological/chemical domain.
3. Furthermore, as shown in Tab. 1 and Fig. 3, the empirical performance seems not significant compared with baselines, and many generated drugs are not realistic molecules. The major improvement is in diversity metric. However, I would say higher diversity with lower quality actually indicates lower generative modeling capacity. I suggest the authors to further improve the model's empirical performance.
4. Some important baseline is missing [1].

[1] Meng Liu, Youzhi Luo, Kanji Uchino, Koji Maruhashi, and Shuiwang Ji. Generating 3d molecules for target protein binding. ICML 2022
for target protein binding.

**Summary Of The Paper:**

The paper studied structure-based protein design, which means generating drugs specifically for a given protein target. The author proposed a conditional diffusion model for this task, which is an extension of the previous equivariant diffusion model [1] for 3D molecule generation. The author considered both conditional generation and inpainting generation tasks. Experiments show that the model can match the performance of current other methods.

[1] Emiel Hoogeboom, Vıctor Garcia Satorras, Clement Vignac, and Max Welling. Equivariant diffusion for molecule generation in 3d. In International Conference on Machine Learning, pp. 8867–8887. PMLR, 2022.


**Summary Of The Review:**

The idea of diffusion models for structure-based drug design (SBDD) is good, but both the SBDD task and the equivariant diffusion method are not supervising things for ML community. Besides, the adoption of the "inpainting" technique from image generation is not clearly natural and useful for this problem. Besides, the empirical performance is also not significant, and some important baseline is missed.

---

> ### Author Response · Authors · 2022-11-20
> **Response to reviewer bBKg**
>
> **Q1: Though the first diffusion for this specific task, the method itself of equivariant diffusion is not new. …. Thus the papers seem to be a little incremental for ML conferences, with limited technical contributions.**
>
> We thank the reviewer for this comment. Concerns about lack of novelty are addressed in our general response.
>
> Even though equivariant diffusion models have been prevailing very recently, the majority of the work focuses on molecular conformation generation, which aims to generate molecular conformations given molecular structures. We argue that structure-based drug design is different as it requires generating both molecular structures and conformations given conditioning protein pockets.
>
> **Q2: Besides, the task has also been well formulated in previous ML papers.**
>
> We agree that the task of structure-based drug design has been well formulated before, however, this does not mean that it is now impossible to make contributions to this field. If anything, substantially more work is needed by the ML community to make SBDD with AI ever have any impact in the real world. We would argue that measuring progress on well-formulated tasks should be encouraged within the ML community and ICLR.
>
> **Q3: For the proposed method, the author also discussed the "inpainting" scenario. However, it's very unclear to me how this could be useful? It feels like a direct and unnatural adoption of diffusion models on images, but in my mind perturbing the protein targets is not very meaningful in the biological/chemical domain.**
>
> We apologize if the motivation was not made clear enough in the manuscript. We believe that the inpainting idea quite naturally applies to target-specific small molecule design. We view the biological system of protein-ligand complexes as a complete data point. Masking out the ligand and filling it back in is therefore comparable to inpainting of missing regions in images. Making this connection, we can utilize methods that have been originally developed in the image domain and already proved their efficiency.
>
> During training, we model the joint distribution of the protein target and the ligand. During sampling, we mask out the ligand and the protein target still serves as context for the generative model. The key difference to direct conditioning is that inpainting uses a noised representation of the pocket (using the true diffusion process) to align the representation with the training procedure where the joint distribution is modeled.
>
> **Q4: Furthermore, as shown in Tab. 1 and Fig. 3, the empirical performance seems not significant compared with baselines, and many generated drugs are not realistic molecules. The major improvement is in diversity metric. However, I would say higher diversity with lower quality actually indicates lower generative modeling capacity. I suggest the authors to further improve the model's empirical performance.**
>
> Thank you for pointing this out. We agree that diversity should not be increased at the cost of quality but do not believe that the presented data suggest that this is happening, considering how noisy the statistics (large standard deviations) and yet how close the values for different methods are.
>
> We re-iterate in our general response about our main contribution and we use the probabilistic framework to tackle drug discovery from a more holistic perspective rather than narrowly optimizing a single property. We also show that we could optimize certain properties further with an evolutionary algorithm (Section 4.5), something not possible with previous work.
>
> **Q5: Some important baseline is missing [1].**
>
> We thank the reviewer for this comment. We are currently working on additional baselines but could not finish the experiments in time given the short time period (first half of the rebuttal period). In particular, we are working on retraining the Pocket2Mol and 3D-SBDD models with the Binding MOAD datasets and various ablation studies. We will also work on the comparison with GraphBP.
>
> **Q6: Quality is high but can still be improved. E.g., There is an anonymous reference above Equation.10. However, even the title of the anonymous reference is missed, making the sentence a little confusing...**
>
> We thank the reviewer for pointing this out we have fixed the reference.

---

### Official Review · Reviewer_WRoN · 2022-10-21

**Confidence:** 3
**Correctness:** 2
**Technical Novelty And Significance:** 2
**Empirical Novelty And Significance:** Not applicable
**Recommendation:** 3

**Clarity, Quality, Novelty And Reproducibility:**

* The paper is clear and easy to understand.
* Novelty is limited as stated above.

**Strength And Weaknesses:**


[Strength]
* For the first time, they approach the 3D molecule generation problem as an inpainting problem, which is novel.
* They evaluate 3D molecule generation methods on a new benchmark to reflect more realistic binding scenarios.
* The paper is easy to read and well organized.
* Domain-specific case studies show various aspects of the model.


[Weaknesses]
* Although their approach is novel, the core model relies on the existing model EGNN with minor modifications.
* The primary goal of molecule generation is to achieve high affinities (low VINA scores). However, the proposed method underperforms not only other baselines but also the test set. Therefore, the motivation of the proposed methods is not convincing.


**Summary Of The Paper:**

This paper proposes a diffusion model that conditionally generates 3D molecules. The proposed model modifies the existing E(n)-equivariant Graph Neural Networks with a new coordinate update step considering the characteristics of pocket coordinates. They frame pocket conditioning in two ways; pocket-conditioned small molecule generation and joint distribution with inpainting. Besides the proposed model, they also curate a new benchmark reflecting more realistic binding scenarios. Experiments show that the proposed method is better than other baselines in terms of diversity.



**Summary Of The Review:**

Since it does not outperform others and the novelty is marginal it's straightforward to score it as rejected.

---

> ### Author Response · Authors · 2022-11-20
> **Response to reviewer WRoN**
>
> **Q1: Although their approach is novel, the core model relies on the existing model EGNN with minor modifications.**
>
> We thank the reviewer for highlighting the novelty of our approach. To achieve E(3)-equivariance, we adapted the EGNN model for our purposes. As pointed out by Reviewer Czkk, E(3)-equivariance is not suitable for this task. In the updated manuscript, we propose to modify the EGNN model to achieve SE(3)-equivariance (remove reflection compared to E(3)) in Section 3.1 and provide a proof in Appendix F. However, we would like to clarify it was not our goal to develop a new equivariant graph neural network.
>
> We would also like to reiterate our contribution that we formulate structure-based drug design (SBDD), which is a key problem in drug discovery, as a conditional generation problem and propose to solve it with diffusion models which eliminates the necessity to generate atom by atom. Further comments on the novelty are summarized in our general response.
>
> **Q2: The primary goal of molecule generation is to achieve high affinities (low VINA scores). However, the proposed method underperforms not only other baselines but also the test set. Therefore, the motivation of the proposed methods is not convincing.**
>
> We acknowledge the reviewer's concern but would argue that saying achieving high affinities is the primary goal of molecular design is an incomplete representation of the long and complicated multi-objective optimisation process that is drug discovery which includes toxicity, absorption and metabolic stability. Boiling down this multi-dimensional task to a single metric is not representative of the problem. Furthermore, hit-identification (finding high affinity compounds) is one of the easiest stages of drug discovery. It is substantially harder to optimize these high affinity molecules for the properties mentioned above to become a suitable drug [1]. The baseline methods you are referring to do not attempt to perform this beyond pure distribution learning, whereas our framework is flexible enough to be used for both early stage molecular design and optimization of existing high affinity molecules.
> As mentioned in the general response, we use the probabilistic framework to tackle drug discovery from a more holistic perspective rather than narrowly optimizing a single property. The key advantage of this approach is that optimization of desired properties is delegated to the dataset curation step, in which implicit biases are easier to incorporate.
>
> [1] Hoffer, L., Muller, C., Roche, P. and Morelli, X., 2018. Chemistry‐driven Hit‐to‐lead Optimization Guided by Structure‐based Approaches. Molecular Informatics, 37(9-10), p.1800059.

---

### Official Review · Reviewer_Czkk · 2022-10-24

**Confidence:** 4
**Correctness:** 2
**Technical Novelty And Significance:** 2
**Empirical Novelty And Significance:** 2
**Recommendation:** 3

**Clarity, Quality, Novelty And Reproducibility:**

The work is a close analogue of pocket2mol using ideas from Hoogeboom. Unfortunately, it does not appear to work as well as prior work, and it does not seem to add any productive ideas, nor discuss in any clarity the lack of success of some of the interesting ideas, e.g. the inpainting strategy, which appears to have scored worse overall than conditional generation.  The authors provide a new dataset for small molecule generation tasks, however, they don't evaluate prior related codes on this dataset.  The authors provide their code and their datasets, so I expect (though I haven't tested) that the work is reproducible.


**Strength And Weaknesses:**


The paper has a couple of strengths: the task that it attacks is of high importance for human health and the ideas that it discusses represent modern and powerful developments in machine learning.  The paper sets out to test inpainting in the context of small molecule generation in a pocket, a technique that worked extremely well for protein generation in David Baker's lab.  Importantly, the paper also adds a new protein-ligand dataset for generative small-molecule tasks.

The weaknesses of this work overwhelm the positives, unfortunately.  Although the particular task of generating desirable ligands that match a pocket is noble, there is no clear argument why the direct generation would be desirable in principle (for example, it is possible that a more practical approach is to simply learn a surrogate function.) Of course, this would not be important if the results were overwhelming; however, they are not. The model performs significantly worse than pocket2mol.  Importantly, the authors do not even evaluate pocket2mol on the new dataset that they generated (even though the code for pocket2mol is publically available).

Also the paper spends a lot of time discussing parts unrelated to this work (the paper does not use eq (3), (4), (5); instead it predicts the noise, as per eq (6)). Similar in page 5, the paper spends a lot of space discussing things that are not used.  Perhaps some of these things could move to an appendix or a different publication.  In page (4) what is the reference to Anonymous. ICLR2023 submission. 2022?

Importantly, the authors discuss an E3 architecture but then mention rigid body transformations, which is SE3 only (E3 without reflections).  If they indeed use E3 (as appears to be the case, though I didn't check the code), then it is possible that update (10) might perhaps break due to a reflection symmetry of E3.  Did the authors check that indeed there is no mixup of stereo chemistries in the generated molecules (the distinction is important because a reflection could turn a therapeutic drug to a killing toxin.)

Why did the inpainting idea not work as well as expected based on the prior work by Baker's lab?  Also, why did the authors suggest that the inpainting was competitive to the conditional generation (it was possibly better on only 1 of 12 subtasks in tables 1, 2).  Why does the model perform so much worse than pocket2mol on the common CrossDocket set?  Why didn't the authors also run pocket2mol (code is available) on their new MOAD binding dataset?


**Summary Of The Paper:**

The paper presents a diffusion-based model for generating a small molecule complete with 3D atomic coordinates in a protein pocket.  The performance of the model on a prior benchmark is well below that of pocket2mol (a related model) on the CrossDocket test set. The paper also presents a new dataset for the same task, which the authors call "binding MOAD" (from Mother Of All Databases).

**Summary Of The Review:**

This method is valid, but not strong enough for ICLR2023.  The paper uses space on unnecessary items and avoids trying and to understand some of the results, which could have strengthened the presentation or lead to an improvement.  The current version of this paper is not at the level of ICLR2023.

---

> ### Author Response · Authors · 2022-11-19
> **Response to reviewer Czkk (1/2)**
>
> **Q1: The weaknesses of this work overwhelm the positives, unfortunately. Although the particular task of generating desirable ligands that match a pocket is noble, there is no clear argument why the direct generation would be desirable in principle (for example, it is possible that a more practical approach is to simply learn a surrogate function.) Of course, this would not be important if the results were overwhelming; however, they are not. The model performs significantly worse than pocket2mol. Importantly, the authors do not even evaluate pocket2mol on the new dataset that they generated (even though the code for pocket2mol is publically available).**
>
> We thank the reviewer for this valuable comment. We discuss this in our general response:
>
> We believe that the interpretation of the reported results is non-trivial and our goal is not to optimise a particular score. Even though high scores are desirable, of course, we do not expect any of the presented methods to significantly deviate from the scores observed on the real data the models have been trained on. This is especially true for a method like DiffSBDD that directly attempts to approximate the data distribution. By reporting these numbers, we hope to convince readers that the model is functioning and captures important aspects of the data distribution. The reported metrics are basic approximations of some desired molecular properties and only cover a small part of what will ultimately make a successful drug. If the generative model manages to perfectly match the data distribution it might capture other properties that are not reflected in these computational metrics at all (which would be the ideal scenario).
> Nevertheless we added a small example in Section 4.5 that shows how the model can be extended with an evolutionary algorithm to improve a score of interest.
>
> **Q2: Also the paper spends a lot of time discussing parts unrelated to this work (the paper does not use eq (3), (4), (5); instead it predicts the noise, as per eq (6)). Similar in page 5, the paper spends a lot of space discussing things that are not used. Perhaps some of these things could move to an appendix or a different publication. In page (4) what is the reference to Anonymous. ICLR2023 submission. 2022?**
>
> We thank the reviewer for this suggestion. We moved these discussions to the appendix and updated the corrupted reference.
>
> **Q3: Importantly, the authors discuss an E3 architecture but then mention rigid body transformations, which is SE3 only (E3 without reflections). If they indeed use E3 (as appears to be the case, though I didn't check the code), then it is possible that update (10) might perhaps break due to a reflection symmetry of E3. Did the authors check that indeed there is no mixup of stereo chemistries in the generated molecules (the distinction is important because a reflection could turn a therapeutic drug to a killing toxin.)**
>
> We thank the reviewer for this insightful comment. Indeed, the generative process as presented in the original draft is E(3)-equivariant and thus insensitive to reflections. In the updated draft, we propose a simple modification of the EGNN layer that breaks reflection symmetry and enables the model to distinguish chiral molecules in Section 3.1 (We also provide the proof in Appendix F). We re-trained all models accordingly and updated the results. Unfortunately, the CrossDocked models could not be fully trained within the first half of the rebuttal period because of the long training times (with more data than BindingMOAD). Hence, all results reported for CrossDocked are still for the E(3)-model. We will provide updates as soon as they are available.
>
> **Q4: Why did the inpainting idea not work as well as expected based on the prior work by Baker's lab? Also, why did the authors suggest that the inpainting was competitive to the conditional generation (it was possibly better on only 1 of 12 subtasks in tables 1, 2). Why does the model perform so much worse than pocket2mol on the common CrossDocket set? Why didn't the authors also run pocket2mol (code is available) on their new MOAD binding dataset?**
>
> We thank the reviewer for this comment. All results of the inpainting methods are virtually on par with the direct conditional method when the same pocket representation ($C_\alpha$) was used, especially when we also consider the standard deviations. Similarly, the scores of competing methods are usually very close to those of DiffSBDD with the full-atom pocket representation when considering the standard deviation (except for synthetic accessibility). For the reasons described above, we do not expect to get scores that deviate a lot from those calculated for reference ligands.

---

> > ### Author Response · Authors · 2022-11-19
> > **Response to reviewer Czkk (2/2)**
> >
> > **Q5: The work is a close analogue of pocket2mol using ideas from Hoogeboom. Unfortunately, it does not appear to work as well as prior work, and it does not seem to add any productive ideas, nor discuss in any clarity the lack of success of some of the interesting ideas, e.g. the inpainting strategy, which appears to have scored worse overall than conditional generation. The authors provide a new dataset for small molecule generation tasks, however, they don't evaluate prior related codes on this dataset.**
> >
> > We thank the reviewer for the comment. But unfortunately, we cannot agree with the reviewer on this comment. Previous work, such as Pocket2Mol, even though demonstrated good empirical performance, relied on the strong assumption to generate molecules atom by atom and also took longer to sample. We eliminated the assumption and first introduced diffusion models (which have been shown to be successful in a variety of 3D molecular generation problems such as conformation generation [1], antibody design [2], protein design [3]). We argue that the inpainting approach is powerful for this task because of its increased versatility (learning the joint distribution vs. learning the conditional distribution) with no notable performance degradation given the same target protein representation.
> > Given the short time period (first half of the rebuttal period), we could not finish the experiment to run the baseline methods on the new Binding MOAD dataset, but we will update the result as soon as it finishes.
> >
> > [1] Jing, B., Corso, G., Chang, J., Barzilay, R. and Jaakkola, T., 2022. Torsional Diffusion for Molecular Conformer Generation. arXiv preprint arXiv:2206.01729.
> >
> > [2] Luo, S., Su, Y., Peng, X., Wang, S., Peng, J. and Ma, J., 2022. Antigen-specific antibody design and optimization with diffusion-based generative models. bioRxiv.
> >
> > [3] Anand, N. and Achim, T., 2022. Protein Structure and Sequence Generation with Equivariant Denoising Diffusion Probabilistic Models. arXiv preprint arXiv:2205.15019.

---

### Official Review · Reviewer_qoqG · 2022-10-25

**Confidence:** 4
**Correctness:** 3
**Technical Novelty And Significance:** 3
**Empirical Novelty And Significance:** 1
**Recommendation:** 3

**Clarity, Quality, Novelty And Reproducibility:**

The code was well-documented and easy to run. The results were also available for evaluation. The paper was reasonably clear however the novelty of the proposed DDPM formulation was lacking.

The code provided does not appear to perform molecule standardization/minimization following molecule generation (outside of explicit errors thrown by RDKit due to valence/hybridization errors), is there a rationale for not performing this cleanup step?

**Strength And Weaknesses:**

**Strengths**:

The paper takes a recently published DDPM formulation for modeling protein-ligand interactions and introduces what appears to be a novel approach to 3D pocket conditioning (conditional and inpainting). They also use an experimental database (Binding MOAD) for training the model instead of synthetic data that often has no grounding in reality.

**Weaknesses**:

A fundamental flaw in this work appears to be a misunderstanding by the authors revealed in the second sentence of the abstract. SBDD pipelines start with ligand and protein preparation which is a prerequisite for large-scale docking. This study excludes this extremely important step as ionization/tautomeric states for ligands and H-bond optimization/protonation for proteins are neglected. SBDD is only as good as the structures you put into it (garbage in~garbage out).

The novelty of the proposed DDPM formulation is lacking as this work closely follows the framework developed by Hoogeboom et al. 2022. In addition, the paper does not include results for the inpainting experiments using all-atom protein level graphs. The poor results achieved using only C*α* atoms for the protein are not surprising as SBDD requires explicit representation of rotamers for amino acids as the orientation/protonation of these residues can dramatically alter what interactions can be made and ultimately whether a molecule will bind with sufficient potency to warrant detection in an in vitro assay.

Some of the molecules generated by this approach are not physically possible to produce and would be incredibly unstable. The paper uses a reference to Ertl & Schuffenhauer 2009 to justify that the low SA score may be because it is based on historical knowledge of previously synthesized molecules, but this is precisely the knowledge needed to understand what is possible on planet earth at 1 ATM with the tools we have at our disposal to synthesize real molecules that we can test experimentally to evaluate these predictions. I take a similar issue with the use of the CrossDocked dataset for training as these are in silico docking predictions that have not been experimentally validated and therefore should not be expected to generate results that are grounded in reality.

The paper does not propose a null model to evaluate the significance of the reported results.

**Summary Of The Paper:**

The authors propose DiffSBDD, an approach to structure-based drug design (SBDD) that aims to generate molecules in 3D that are conditioned on a protein binding interface. The approach is E(3)-equivariant and leverages the denoising diffusion probabilistic model (DDPM) formulation inspired by non-equilibrium thermodynamics.

Two distinct approaches to 3D pocket conditioning are proposed. The first keeps the pocket constrained in each denoising step (conditional) and the second approximates the joint distribution of ligand-pocket pairs enabling sampling of both ligand and protein positions (inpainting).

To demonstrate the applicability of the proposed model, the authors create a new dataset of experimentally derived protein-ligand complex data from the publically available database Binding MOAD in addition to a synthetic cross-docked dataset.
De-novo molecule generation was evaluated for the conditional approach in the context of both all-atom and Cα (for protein) level graphs. For the inpainting approach, only Cα level graphs could be evaluated due to computational limitations. In silico metrics were calculated for generated molecules and compared with two recent deep-learning approaches to SBDD (3D-SBDD and Pocket2Mol). **DiffSBDD did not perform better**.

**Summary Of The Review:**

The authors propose DiffSBDD, an E(3)-equivariant 3D-conditional diffusion model for SBDD which they evaluate in the molecular generation of novel/diverse ligands that maximize predicted affinity. While they demonstrate that inpainting can achieve competitive results to other deep learning approaches to SBDD, it was not superior, and no null model is presented to demonstrate the significance of the results. In addition, the curation and preparation of the protein and ligand structures were not reported/neglected calling into question the validity of the results. Lastly, the proposed DDPM formulation does not appear to be a novel contribution which limits the significance of the manuscript contributions. Therefore, my recommendation is that the manuscript is premature for acceptance to ICML.

---

> ### Author Response · Authors · 2022-11-19
> **Response to reviewer qoqG (1/2)**
>
> **Q1: A fundamental flaw in this work appears to be a misunderstanding by the authors revealed in the second sentence of the abstract. SBDD pipelines start with ligand and protein preparation which is a prerequisite for large-scale docking.**
>
> We thank the reviewer for this comment and acknowledge that receptor preparation is indeed a vital stage of SBDD, however, it was not mentioned in the abstract as we have tried to outline the problem as it is commonly presented at other ML conferences [1,2]. We would argue the second sentence of an abstract is to frame the problem domain in broad strokes, in order to make the text accessible to a wide audience, rather than mention a specific sub-task within the traditional SBDD pipeline that the ML audience might not be familiar with.  Furthermore, ligand preparation is not a problem we have attempted to solve with this work.
>
> **Q2: This study excludes this extremely important step as ionization/tautomeric states for ligands and H-bond optimization/protonation for proteins are neglected. SBDD is only as good as the structures you put into it (garbage in~garbage out).**
>
> Providing more chemical context as input to computational SBDD approaches is certainly a promising direction, and the superior performance of the full-atom model on the reported metrics supports this intuition as well. We believe however that these effects are slightly less important in the proposed probabilistic framework (as compared to energy-based methods for example) because recurring patterns in the data can be modeled even without a complete physical model.
> Furthermore, the authors completely agree about the important nature of ionization/tautomeric states and protonation for modeling protein-ligand interactions. However, we would also like to point out that this is a much larger problem in machine learning for structural biology, the solution to which was outside of the scope and contributions of this work. Indeed, many well-published works in the field of AI for structural biology [3, 4] and specifically protein-drug interactions [1, 2, 5] choose to ignore these important factors too. Hence, we felt that it was an appropriate assumption to make in this work as we do not see it as one of our main contributions.
>
> **Q3: The novelty of the proposed DDPM formulation is lacking as this work closely follows the framework developed by Hoogeboom et al. 2022.**
>
> We hope that our work showcases the applicability of diffusion models for a highly relevant scientific problem through a confluence of existing ideas from the fields of denoising diffusion models and equivariant neural network architectures for molecular systems. A more detailed discussion is provided in our general response.
>
> **Q4: In addition, the paper does not include results for the inpainting experiments using all-atom protein level graphs. The poor results achieved using only Cα atoms for the protein are not surprising as SBDD requires explicit representation of rotamers for amino acids as the orientation/protonation of these residues can dramatically alter what interactions can be made and ultimately whether a molecule will bind with sufficient potency to warrant detection in an in vitro assay.**
>
> We agree with the reviewer that the discrepancy between the two pocket representations is not surprising and yet deemed sharing the empirical evidence for it useful. Similarly, we believe that our study of the inpainting approach in the context of conditional molecule generation can be of value to others even without extending it to the all-atom scenario due to memory issues as mentioned in Section 4.2 (the inpainting approach initializes the protein and ligand at the origin and creates fully-connected graphs because of the distance threshold).
>
> **Q5: Some of the molecules generated by this approach are not physically possible to produce and would be incredibly unstable. The paper uses a reference to Ertl & Schuffenhauer 2009 to justify that the low SA score may be because it is based on historical knowledge of previously synthesized molecules, but this is precisely the knowledge needed to understand what is possible on planet earth at 1 ATM with the tools we have at our disposal to synthesize real molecules that we can test experimentally to evaluate these predictions.**
>
> We thank the reviewer for this valuable comment. While we tried to not cherry-pick the presented molecules too much apart from sorting by in silico docking score for Figures 3 and 4, it is true that many of the generated molecules are not viable candidates for experimental testing. The real strength of the method lies in its ability to rapidly generate diverse sets of molecules that can be further filtered and optimized. We added an example (Section 4.5) to show how the model can be utilized in a straightforward way to improve properties of interest in a targeted way.
>
> We completely agree that our justification of the low SA scores was misleading and changed the wording.

---

> > ### Author Response · Authors · 2022-11-19
> > **Response to reviewer qoqG (2/2)**
> >
> > **Q6: I take a similar issue with the use of the CrossDocked dataset for training as these are in silico docking predictions that have not been experimentally validated and therefore should not be expected to generate results that are grounded in reality.**
> >
> > We also agree with the reviewer‘s comment that CrossDocked is less than ideal due to its synthetic nature, however, it is one of the most popular datasets for related models so we used it here in order to provide a fair comparison to other work. We would like to highlight that we have addressed this issue directly by curating and benchmarking on the experimental Binding MOAD dataset as well.
> >
> > **Q7: The paper does not propose a null model to evaluate the significance of the reported results.**
> >
> > We thank the reviewer for this comment. We kindly ask the reviewer to clarify what they mean by a null model in this case. We agree that an ablation study would be useful to determine the effect pocket conditioning has on target molecule generation. If accepted, we promise this to be available in the camera-ready version.
> >
> > **Q8: The code provided does not appear to perform molecule standardization/minimization following molecule generation (outside of explicit errors thrown by RDKit due to valence/hybridization errors), is there a rationale for not performing this cleanup step?**
> >
> > We thank the reviewer for this comment. We did post-process generated molecules slightly (as described in Appendix C), e.g. with a few rounds of force field minimization, following previous methods [1, 2]. We refrained from additional clean-up/filtering steps to better demonstrate the capabilities of the generative neural network itself. In a real-life drug discovery campaign, additional post-processing steps should be considered and the pipeline can be easily extended accordingly.
> >
> > The code to post-process generated molecules is also provided, under the folder analysis/molecule_builder.py process_molecule function.
> >
> > [1] Luo, S., Guan, J., Ma, J. and Peng, J., 2021. A 3D generative model for structure-based drug design. Advances in Neural Information Processing Systems, 34, pp.6229-6239.
> >
> > [2] Peng, X., Luo, S., Guan, J., Xie, Q., Peng, J. and Ma, J., 2022. Pocket2Mol: Efficient Molecular Sampling Based on 3D Protein Pockets. arXiv preprint arXiv:2205.07249.
> >
> > [3] Jumper, John, et al. "Highly accurate protein structure prediction with AlphaFold." Nature 596.7873 (2021): 583-589.
> >
> > [4] Baek, Minkyung, et al. "Accurate prediction of protein structures and interactions using a three-track neural network." Science 373.6557 (2021): 871-876.
> >
> > [5] Stärk, Hannes, et al. "Equibind: Geometric deep learning for drug binding structure prediction." International Conference on Machine Learning. PMLR, 2022.

---

### Author Response · Authors · 2022-11-19
**General Response (1/2)**

We would like to thank all the reviewers for their extensive reviews and valuable comments. We have incorporated many of these changes into the revised manuscript (marked in blue). We are encouraged that many of the reviewers found our proposed method novel and interesting and the curation of experimentally-resolved protein-ligand structures to be a good contribution to the field of 3D-conditioned molecular design. However, common concerns about the work are related to the performance of our model with regard to particular metrics, novelty of the work and the use of the CrossDocked dataset.

In order to address reviewers' concerns about the performance and usefulness, we have included an extra experiment to showcase the power and flexibility of the diffusion model paradigm in SBDD. While being able to elaborate from the candidate ligands is useful on its own, we demonstrate that we can combine this noising/denoising paradigm with a simple evolutionary algorithm to perform powerful molecular optimisation for an arbitrary property (e.g. binding affinity, QED etc).

As a summary response, we address other main points below:

**DiffSBDD did not outperform baseline models**

We believe that the interpretation of the reported results is non-trivial and the goal of the general study was not to optimize a particular score. Even though high scores are desirable, of course, we do not expect any of the presented methods to significantly deviate from the scores observed on the real data the models have been trained on. This is especially true for generative models that directly attempt to approximate the data distribution. By reporting these numbers, we hope to convince readers that the model is functioning and captures important aspects of the data distribution. The reported metrics are basic approximations of some desired molecular properties and only cover a small part of what will ultimately make a successful drug. If the generative model manages to perfectly match the data distribution it might capture other properties that are not reflected in these computational metrics at all (which would be the ideal scenario).
Nevertheless we added a small example in Section 4.5 that shows how the model can be extended with an evolutionary algorithm to improve a score of interest.

**Unrealistic molecules**

Several reviewers have correctly pointed out that some of the generated molecules are not realistic. We completely agree and would like to stress that we wanted to present both high-quality examples as well as failure cases in the main results in order to be completely transparent about our work. Indeed the quality of our generated molecules varies substantially with the target chosen (as shown in Appendix Figure 13) and we have tried to not only pick high-performing examples. As an example of high performance, we would like to point the reviewer to the generated molecules in Figure 4 which were not curated in any way. If the reviewers believe the work would be improved by showing more high-quality examples in the main text we would be happy to do so.

**Novelty of the work**

We fully acknowledge that our work has built on findings from the Hoogeboom et al. paper [1], however, we argue that our major contributions lie in the rigorous equivariance considerations for the 3D-conditional generative setting, which has not been addressed by the previous work. Tackling structure-based drug design with denoising diffusion probabilistic models is a new, flexible and powerful paradigm, with which we have provided extensive experimental validation and interpretation. We demonstrate this by means of two distinct approaches to target-specific molecule generation, direct conditioning and inpainting, and the newly added case study for out-of-the-box molecule optimization with an evolutionary algorithm. We believe that the core novelty of our paper lies in the confluence and execution of these ideas for SBDD, thereby providing a new perspective on the problem.

We furthermore hope that our analysis of experimentally determined structures from the Binding MOAD database brings much needed attention to more realistic datasets for computational molecular design.

---

> ### Author Response · Authors · 2022-11-19
> **General Response (2/2)**
>
> **Usefulness of the inpainting method**
>
> We have further investigated the inpainting approach and discovered inconsistency issues with the straightforward “replacement method” similar to previously reported findings [3, 4] that occasionally lead to generation of displaced ligands. Our new results show that a simple resampling strategy [4] can mitigate this problem consistently without increasing the computational burden.
>
> **Use of the CrossDocked dataset**
>
> Reviewer qoqG raised the valid issue of using the synthetic CrossDocked dataset and the effect this may have on the realism of the generated samples. We entirely agree with this sentiment and it is why we have also used the experimental Binding MOAD dataset. However, CrossDocked is currently the standard dataset used for this task so we felt it was appropriate to include here as a means of fair comparisons to previous work.
>
> **Changes made**
>
> We thank the reviewers for their many excellent suggestions on how to improve the manuscript. The summary of minor changes is as follows:
>
> * We added a simple evolutionary algorithm to demonstrate it could be used to optimize the molecular property of interest, in Sec. 4.5, shown by Figure 5.
> * As suggested by Reviewer Czkk, we proposed a modified version of EGNN which was originally E(3) equivariant to be SE(3) equivariant. We proved SE(3) equivariance and demonstrated its effectiveness empirically with a chirality classification task [1]. All Binding MOAD experiments have already been repeated with this new model and performance metrics have been updated accordingly. Unfortunately, we did not manage to retrain all models on the CrossDocked dataset before the paper revision deadline but we will provide updates during the second discussion stage.
> * We added the experimental results of the conditional setting of DiffSBDD on Binding MOAD with the full atom representation.
> * We show that the inpainting method can be consistently improved with resampling (Appendix C.1)
> * We demonstrated that the QuickVina docking score is highly correlated with ligand size (Appendix Figure 6) and updated the sampling procedure to correct for this effect. Again, this change has only been evaluated on the Binding MOAD dataset so far but results for CrossDocked will be communicated during the second discussion stage.
> * We improved the clarity of the paper by moving less relevant items to the appendix following the suggestion by reviewer Czkk.
>
> [1] Hoogeboom, Emiel, et al. "Equivariant diffusion for molecule generation in 3d." International Conference on Machine Learning. PMLR, 2022.
>
> [2] Adams, K., Pattanaik, L. and Coley, C.W., 2021. Learning 3D Representations of Molecular Chirality with Invariance to Bond Rotations. arXiv preprint arXiv:2110.04383.
>
> [3] Trippe, Brian L., et al. "Diffusion probabilistic modeling of protein backbones in 3D for the motif-scaffolding problem." arXiv preprint arXiv:2206.04119 (2022).
>
> [4] Lugmayr, Andreas, et al. "Repaint: Inpainting using denoising diffusion probabilistic models." Proceedings of the IEEE/CVF Conference on Computer Vision and Pattern Recognition. 2022.

---

### Decision · Program_Chairs · 2023-01-20

**Decision:**

Reject

**Justification For Why Not Higher Score:**

All four reviewers raised major concerns on many aspects of this work, and these concerns remain after rebuttals.

**Justification For Why Not Lower Score:**

N/A

**Metareview: Summary, Strengths And Weaknesses:**

This work proposes a diffusion model for structured based drug design. All four reviewers raised major concerns on many aspects of this work, and these concerns remain after rebuttals. Given the consistent concerns, a reject is appropriate.